

# Riding the wave of innovation: immunoinformatics in fish disease control

Siti Aisyah Razali[1,2], Mohd Shahir Shamsir[3], Nur Farahin Ishak[1], Chen-Fei Low[4] and Wan-Atirah Azemin[5]

[1] Faculty of Science and Marine Environment, Universiti Malaysia Terengganu, Kuala Nerus, Terengganu, Malaysia
[2] Biological Security and Sustainability Research Interest Group (BIOSES), Universiti Malaysia Terengganu, Kuala Nerus, Terengganu, Malaysia
[3] Department of Biosciences, Faculty of Science, Universiti Teknologi Malaysia, Skudai, Johor, Malaysia
[4] Institute of Systems Biology (INBIOSIS), Universiti Kebangsaan Malaysia, Bangi, Selangor, Malaysia
[5] School of Biological Sciences, Universiti Sains Malaysia, Minden, Pulau Pinang, Malaysia

Corresponding authors
Siti Aisyah Razali,
aisyarazali@umt.edu.my
Wan-Atirah Azemin,
wanatirah@usm.my

## ABSTRACT

The spread of infectious illnesses has been a significant factor restricting aquaculture production. To maximise aquatic animal health, vaccination tactics are very successful and cost-efficient for protecting fish and aquaculture animals against many disease pathogens. However, due to the increasing number of immunological cases and their complexity, it is impossible to manage, analyse, visualise, and interpret such data without the assistance of advanced computational techniques. Hence, the use of immunoinformatics tools is crucial, as they not only facilitate the management of massive amounts of data but also greatly contribute to the creation of fresh hypotheses regarding immune responses. In recent years, advances in biotechnology and immunoinformatics have opened up new research avenues for generating novel vaccines and enhancing existing vaccinations against outbreaks of infectious illnesses, thereby reducing aquaculture losses. This review focuses on understanding *in silico* epitope-based vaccine design, the creation of multi-epitope vaccines, the molecular interaction of immunogenic vaccines, and the application of immunoinformatics in fish disease based on the frequency of their application and reliable results. It is believed that it can bridge the gap between experimental and computational approaches and reduce the need for experimental research, so that only wet laboratory testing integrated with in silico techniques may yield highly promising results and be useful for the development of vaccines for fish.

## INTRODUCTION

Millions of farmers, food processors, traders, researchers, technical experts, and leaders all over the world are engaged in the daunting challenge of feeding a projected nine billion global population by 2050. Fish and other aquatic products from aquaculture play
a significant role in meeting the dietary needs of all people, as well as the requirements of the poorest for food security (*Mair et al., 2023*). Aquaculture accounts for 49.2% of total aquaculture and fisheries production on a global scale, with proportions varying by region and production sector. Aquaculture is essential to meet the world's need for fish for several reasons, including overfishing, habitat degradation, climate change, pollution, and unsustainable fishing practices (*Divya & Devi, 2023*). However, the sustainable development of the aquaculture sector is hindered by many factors, with the control of infectious diseases being one of the most significant challenges, as fish disease outbreaks have caused enormous economic losses in the aquaculture industry (*Tavares-Dias & Martins, 2017*; *Peterman & Posadas, 2019*; *Fernández Sánchez et al., 2022*; *Abdelrahman et al., 2023*). Losses in fish production, revenue, livelihoods, and international trade (citation) are major components of economic losses caused by fish disease outbreaks in aquaculture, emphasizing the need for effective fish disease management strategies.

Despite antibiotics or chemotherapeutics being used for fish disease treatment in aquaculture, drug resistance issues and safety concerns become obstacles to resolving fish disease outbreaks (*Harikrishnan, Balasundaram & Heo, 2011*; *Sneeringer, Bowman & Clancy, 2019*). Thus, fish vaccinations have been extensively employed in the aquaculture industry. Prior to deployment, fish vaccines, like those used in human and veterinary medicine, must pass stringent tests for safety and efficacy. In safety assessments, potential adverse effects on vaccinated fish, non-target species, and the environment are evaluated. These tests ensure that the vaccine does not cause excessive damage to the fish, has no negative effects on non-target organisms, and does not introduce harmful substances into the environment (*Irshath et al., 2023*). The effectiveness of fish vaccines is evaluated both in the laboratory and in the field. The immune response of the fish is monitored in the laboratory to corroborate that the vaccine induces an adequate immune response. Trials are conducted in the field to ensure that the vaccine provides protection against the targeted pathogen under real-world conditions.

Notably, although vaccines considerably reduce the likelihood of disease outbreaks, they do not guarantee complete immunity. Individual fish may respond differently to vaccination, similar to other animals, due to a variety of factors, including genetic variability, age, nutritional status, stress, and concurrent infections. This is why continuous monitoring of the efficacy and safety of vaccines is necessary (*Zimmermann & Curtis, 2019*). Vaccines are an essential component of the sustainable management of aquaculture, as they contribute to disease control and fish welfare while reducing the use of antibiotics. They provide a proactive and preventative approach to health management, which aligns with the overarching goal of assuring global food security. As is the case with all medications, the key to their successful application rests in their application in accordance with scientific research and established guidelines.

A fish vaccine typically contains a substance derived from pathogenic microorganisms in non-pathogenic forms that act as an antigen. By stimulating the fish's immune system to combat a specific pathogen, the system is permitted to create a response, as well as a "memory" to cause the acceleration of the response, when the specific organism that causes the disease creates future infections (*Yanong, 2017*; *Ma et al., 2019*; *Kayansamruaj, Areechon*

*& Unajak, 2020*). Traditional methods were used to develop a variety of vaccines, including killed whole-cell, live-attenuated, recombinant DNA, subunits, and toxoid vaccines (*Gudding & Van Muiswinkel, 2013*; *Ma et al., 2019*; *Bouazzaoui et al., 2021*). However, most authorized and commercial vaccines currently in use in the aquaculture industry are killed whole-cell vaccines while other vaccine groups are still being studied in live animals or are in the experimental phase (*Adams, 2019*; *Mohd-Aris et al., 2019*).

The killed whole-cell vaccine, also known as bacterin, is among the oldest vaccination technologies indigenously manufactured by many developing countries (*Maeda et al., 2021*). To make such a vaccine, it requires organisms that must be inactivated or that have died through physical or chemical procedures like inactivation with heat, irradiation with UV, or inactivation through formalin or chloroform (*Lee et al., 2012*; *Tafalla, Bøgwald & Dalmo, 2013*). When administered to the host, they induce strong protective humoral immune responses against those pathogens (*Damodharan et al., 2021*). Using killed whole-cell vaccines can prevent a number of viral disease outbreaks, which include infectious necrosis of the pancreas, spleen, and kidney; and pancreatic disease viruses; as well as bacterial diseases such as Vibriosis, enteric septicaemia of catfish, and Streptococcus infections (*Assefa & Abunna, 2018*; *Ma et al., 2019*).

Live-attenuated vaccines are types of vaccines that contain live microorganisms whose virulent properties were disabled under specific cultivation conditions to generate a broad immune response (*Abdelhamed, Lawrence & Karsi, 2018*; *Heckman et al., 2022*). Many scientific studies have focused on these vaccinations, which are being investigated for commercialization as fish vaccines due to their capacity to combat 209 infectious diseases caused by recognised and unknown pathogenic microorganisms still under investigation (*Kayansamruaj, Areechon & Unajak, 2020*). Unlike killed whole-cell vaccines, live-attenuated vaccines are able to induce both cell-mediated and humoral immune responses (*Shoemaker et al., 2009*; *Côté-Gravel, Brouillette & Malouin, 2019*). These vaccines with a minimum dosage are adequate to elicit long-lasting protective immune responses as they mimic the real infections caused by pathogens. This incident preferentially evokes T-cell proliferative responses relative to B-cell responses (*Tajimi et al., 2019*; *Muñoz Atienza, Díaz-Rosales & Tafalla, 2021*). Thus, they confer greater adaptive immune protection in fish compared with the induction of killed whole-cell vaccine or subunit vaccine (*Sudheesh & Cain, 2017*; *Mohd-Aris et al., 2019*). For instance, live-attenuated vaccinations have prevented herpesvirus disease (*Dhar, Manna & Thomas Allnutt, 2014*; *Huang et al., 2021*), columnaris disease (*Shoemaker et al., 2011*; *Cai & Arias, 2021*), and bacterial kidney disease (*Evensen, 2016*; *Delghandi, El-Matbouli & Menanteau-Ledouble, 2020*) caused by pathogens; KHV *Herpesvirus*, *Flavobacterium columnaris*, and *Renibacterium salmoninarum*, respectively.

The recombinant DNA vaccine is one of the experimental vaccines now in use and in research. Using the gene gun technique, the pathogen gene is cloned into the vector before being introduced into the host. Subsequently, the protein that functions as an antigen will be synthesised within the host and will elicit an immunological response (*Lorenzen & LaPatra, 2005*; *Hølvold, Myhr & Dalmo, 2014*; *Collins, Lorenzen & Collet, 2019*). Similar to live-attenuated vaccines, it induces both humoral and cellular immunity (*Nascimento*

& Leite, 2012; Bedekar, Kole & Tripathi, 2020). For instance, Nile tilapia vaccinated with the recombinant DNA vaccine SL7207-pVAX1-sip had a higher survival rate following *Streptococcus agalactiae* infection (Zhu et al., 2017); while flounder fish were conferred with a protective immune response by administering a vaccine based on DNA that encoded the VAA gene of *Vibrio anguillarum* (Xing et al., 2019). This demonstrates that recombinant DNA vaccines are useful tools in investigating the key factor in the pathogenicity of the etiological agent to the fish and are economically viable in animals with extremely high value (Khan et al., 2016; Mzula et al., 2019).

When it is difficult to cultivate an organism, subunit vaccines are advantageous because they utilise the immunogenic component of the organism. Subunit vaccines may incorporate toxoids, subcellular fragments, and surface antigens. In comparison to inactivated, whole-organism vaccinations, these vaccines have limited immunogenicity. To enhance immunogenicity, adjuvants are necessary (Dadar et al., 2017). Many studies have reported that subunit vaccines such as recombinant subunit vaccines of grouper sleepy disease iridovirus (GSDIV) with montanide ISA could be utilised to decrease grouper mortality due to GSDIV infection (Mahardika et al., 2016). In addition, the efficacy of three different subunit vaccines against *Aeromonas salmonicida* infection in rainbow trout *Oncorhynchus mykiss* has been shown to significantly lower mortalities after 3 weeks (Marana et al., 2017). Although subunit vaccinations pose a relatively minimal risk of negative effects, retaining their antigen in their native form during the purification process may be difficult. Thus, organisms may be unable to detect antigens, resulting in these proteins failing to elicit an immune response in the host (Wang, Jiang & Wang, 2016; Abinaya & Viswanathan, 2021).

The composition of fish vaccines may differ from vaccines intended for human use with respect to the adjuvants and preservatives that are suitable for aquatic environments. Human vaccines are formulated with components safe for human use. Due to the differences in fish immune systems, comprehensive research and development are being conducted to formulate adjuvants to enhance subunit vaccines' effectiveness in fish species. Meanwhile, human vaccines are subjected to extensive clinical trials prior to the approval, and the regulatory requirements for fish vaccines are specific to the aquaculture industry. Fish vaccines are calibrated to cater to the unique disease profiles and the needs of particular fish populations to ensure optimal health and protection against diseases.

Immunostimulants or adjuvants are routinely added to vaccinations containing inactivated pathogens or recombinant antigens to serve as vaccine carriers, thereby enhancing the vaccine's efficacy and eliciting a powerful immune response (Tafalla, Bøgwald & Dalmo, 2013; Huang et al., 2014; Munangándu et al., 2020; Guo & Li, 2021). However, its effectiveness depends on the method of administration. There are three methods for administering vaccinations to fish: injection, immersion, and oral immunisation (Ringøet al., 2014; D'Amico et al., 2021). In general, injection is superior to oral delivery and immersion vaccination; however, this preference is dependent on the fish's size (Embregts & Forlenza, 2016; Bøgwald & Dalmo, 2019). Notably, these approaches are only used on healthy fish because they are preventative and not curative (Wali & Balkhi, 2016; Miccoli et al., 2021). In general, the advantages and disadvantages of the common

**Table 1** Advantages and disadvantages of different vaccination methods that are common in fish farming.

| Vaccine administration | Advantages | Disadvantages |
| --- | --- | --- |
| Injection | ● Controlled and precise dosage for optimal immune stimulation, reducing the risk of under or over-dosing.<br>● Higher efficacy due to precise delivery of the vaccine directly into the fish, resulting in a potent immune response, thus, providing longer-lasting immune protection against the target pathogen. | ● Induces stress due to the mechanical handling of fish during vaccination, which could potentially lead to negative physiological responses and reduced immunity.<br>● Labor-intensive and time-consuming to vaccinate large numbers of fish.<br>● Not suitable for small fish. |
| Immersion | ● Non-invasive approach, thus, reducing stress and minimizing the risk of injury during vaccination.<br>● Less labour-intensive, thus, cost-effective for large-scale aquaculture.<br>● Time-efficient for mass vaccination of the cultured species. | Lower efficacy compared to the injection method due to the variations in vaccine uptake by different individuals, thus leading to inconsistent immune responses and protection levels in the vaccinated population of the cultured species. |
| Oral | ● Non-invasive approach, and it is typically well-tolerated by fish, reducing stress and the potential risk of injury during vaccination.<br>● Applicable to vaccinate small fish or fry.<br>● Oral vaccines can be incorporated into fish feed, thus making it more practical for small- to large-scale fish farming. | ● Variable uptake of the vaccines through fish digestive systems leads to inconsistent immune responses and protection levels in the vaccinated population of the cultured species.<br>● Heat-sensitive vaccines could lose their efficacy during feed processing, storage, or digestion. |

methods used to vaccinate fish could be summarized in Table 1. Overall, the selection of vaccination approach depends on various factors that include the fish species, types of vaccine, and the scale of the operation. A combination of different administration approaches or the use of different adjuvants and immunostimulants may be required to optimize the immune response and to ensure vaccine efficacy in disease prevention. Close monitoring is essential to evaluate the health status of the vaccinated fish population to identify potential adverse effects of the vaccines on fish health.

Nonetheless, commercial vaccine development is constrained by cost-effectiveness in the field. In comparison to terrestrial animals, fish require a higher antigen dose, therefore, developing cost-effective inactivated viral vaccines has proven problematic (*Sommerset et al., 2005*; *Muktar & Tesfaye, 2016*; *Shefat, 2018*). For example, live-attenuated vaccines require proper storage since they are live (*Kumru et al., 2014*; *Prosser et al., 2021*; *Pambudi et al., 2022*). Similarly, the killed whole-cell vaccine type involved a high manufacturing cost in cell culture tests, where a significant number of microorganisms are necessary to produce immunity, and the need for multiple injections may also exist depending on the distinguishing qualities of the vaccine (*Vaughn, Whitehead & Durbin, 2009*; *Dias et al., 2013*; *Rodrigues & Plotkin, 2020*). Moreover, when a recombinant DNA vaccine is used as an alternative, immunologic tolerance (hyporesponsive) may develop because the antigen is expressed in the host (*Liu et al., 2011*; *Peignier & Parker, 2020*) which renders the host incapable of mounting an immunological response following vaccination (*Poolman & Borrow, 2011*; *Hobernik & Bros, 2018*; *Brisse et al., 2020*). Furthermore, certain chemical treatments used in killed whole-cell vaccine development such as formaldehyde may alter antigenicity. This alteration necessitated the use of adjuvants in single or repeated doses to lessen the risk of antigenicity. Thus, this phenomenon not only raises production costs
but also formulation and administration complexity (*Furuya et al., 2010*; *Sanders, Koldijk & Schuitemaker, 2015*; *Martínez-Flores et al., 2021*).

Certain fish species are too weak to withstand the stress induced by immunisation and may experience severe side effects after vaccination (*Parra, Reyes-Lopez & Tort, 2015*; *Mugwanya et al., 2022*). Moreover, it is difficult to analyse the relationship between pathogen and vaccine-induced immunity in fish species (*Ángeles Esteban, 2012*; *Biller-Takahashi & Urbinati, 2014*; *Magadan, Sunyer & Boudinot, 2015*; *Smith, Rise & Christian, 2019a*). Moreover, severe issues related to disease might develop at the stages of larvae or fry in other species, that is, prior to the organism growing sufficiently so that vaccination can occur or fully operational immune systems can form (*Dhar, Manna & Thomas Allnutt, 2014*; *Muktar & Tesfaye, 2016*; *Hazreen-Nita et al., 2019*). In this regard, the computational immunology technique, also known as the immunoinformatics approach, is one way that these limitations can be circumvented and overcome. Immunoinformatics bridges the gap between computer science and immunology by employing computational resources and methods to manage and comprehend immunology data. It contributes to the management of large datasets and aids in the creation of new hypotheses regarding immune responses (*Tomar & De, 2014*; *Chatanaka et al., 2022*; *Wong et al., 2022*).

While the practical implementation of immunoinformatics in aquaculture has been established, it is essential to recognise that there are significant differences between fish and human immune systems (*Wang, Chen & Wang, 2019*). Understanding these differences is crucial for the efficient application of immunoinformatics in the development of vaccines and treatments for fish. The innate immune system of fish, which is recognised as the first line of defence against a variety of pathogens, plays a more significant role than its homologue in mammals. Notably, primitive fish species with no jaws, such as lampreys and hagfish, have a profoundly different immune system than jawed vertebrates. This system lacks the typical B and T cells and Major Histocompatibility Complex (MHC) molecules observed in humans and advanced fish species. For adaptive immunity, these jawless species rely on a unique system of variable lymphocyte receptors (VLRs). Jawed fish, including cartilaginous (such as sharks) and bony (such as trout) species, have evolved B and T cells and MHC molecules, heralding the "modern" emergence of adaptive immunity (*Buchmann, 2014*; *Mitchell & Criscitiello, 2020*; *Wang, Chen & Wang, 2019*). Despite this, fish immune systems are less sophisticated than those of mammals, including humans. For instance, fish have fewer subsets of T cells, and their B cells are less diverse. The disparity also extends to MHC molecules. Fewer MHC class II molecules are present in fish than in humans, and their function is not as well understood. Some species, such as the Atlantic cod, lack MHC II molecules but possess a larger number of MHC I molecules (*Boehm, Iwanami & Hess, 2012*). Essentially, these differences highlight the need for predictive models and algorithms that are uniquely tailored to the immune response pathways of fish. Ideally, the data used to train these algorithms should be derived from the immune responses of fish. Taking into consideration these differences will permit the effective application of immunoinformatics to the development of more effective preventative and therapeutic measures for aquaculture health management.
Immunoinformatics has been applied in many research studies, particularly for disease prevention strategies, such as predicting immune cell populations, modelling immune responses, and studying autoimmune disorders and allergies. As our understanding of the immune system has increased in breadth and depth, this approach has naturally evolved to match this progression, giving rise to a term that encompasses this broader spectrum of activities. Immunoinformatics has emerged as a game-changing tool in the development of fish vaccines, addressing major obstacles and accelerating progress. Identification of immune epitopes within fish pathogens is one of the most important applications. Using immunoinformatics, researchers can effectively identify these epitopes: sections of the pathogen's genetic sequences with the potential to elicit an immune response in fish (*Forouharmehr et al., 2022a*; *Islam et al., 2022a*; *Islam et al., 2022b*). This essential knowledge guides the development of vaccines based on epitopes, allowing for targeted immunisation that induces a specific protective immune response. In addition, immunoinformatics contributes substantially to the development of epitope-based vaccines by enabling scientists to select the most immunogenic and conserved epitopes (*Forouharmehr et al., 2022b*; *Islam et al., 2022a*). This strategic approach paves the way for the development of broad-spectrum vaccines, thereby protecting against multiple pathogen strains or variants.

The design of vaccines, the focus of this review, also plays a crucial role in zoonosis prevention. As the interface between humans and animals continues to evolve and become obscure, particularly in the context of aquaculture, the risk of zoonotic diseases those transmitted from animals to humans becomes more pressing. Immunoinformatics can be an indispensable tool for mitigating these hazards and safeguarding public health. The identification of pathogens at an early stage is a crucial application. This technique enables the detection of emergent viral, bacterial, or parasitic strains in fish populations prior to their posing a risk to humans through a combination of genomic sequencing and computational analysis. By identifying and characterising these potential zoonotic hazards in advance, measures can be implemented to prevent their spread and protect human populations. Immunoinformatics facilitates the development of vaccines that can act as barriers to disease transmission at its origin. These vaccines, designed for fish but effective against potential human pathogens, can control the disease in fish populations, thereby reducing the risk of human infection significantly. The same structural understanding and immune system interaction principles can be applied to the design of therapeutic drugs. These prospective treatments could combat pathogens that threaten both fish and human health, thereby serving dual purposes in zoonotic disease control.

This review aims to provide a comprehensive summary of immunoinformatics software that has been used in recent years which is essential for vaccine design, particularly in fish vaccine development. The vaccine design is explicated based on *in silico* epitopes, develop a multi-epitope vaccine, and investigate how immunogenic vaccines interact on a molecular level. Through this approach, the applied and valid results of the use of immunoinformatics to address diseases in fish are examined in relation to how frequently they occur. Additionally, immune mechanisms and immunoinformatics in fish disease are

predicted using TLR signalling pathways and may draw the interest of pharmaceutical and synthetic immunologists in synthesizing and discovering the vaccine's novel potential.

## SURVEY METHODOLOGY

Using Web of Science, Scopus, PubMed, ScienceDirect, and Google Scholar, primary and secondary literature pertinent to this review's topic was evaluated. These databases were used to search for the following terms: "fish diseases" and "aquaculture" in combination with, "immunoinformatics", "computational biotechnology", "bioinformatics", "vaccines", "Epitope prediction", "T cell epitope", "B cell epitopes", "adjuvant", "linker", "Structural modelling", "molecular docking", "molecular dynamics simulation", "immune mechanism", "TLR signalling pathways", "multi-epitope vaccine" along with using "+", "AND", and "OR" for a specific search result. The identified articles were initially examined for relevance to the topic and thoroughly read.

### *In silico* epitope-based vaccine design
### *Epitope prediction*

By preventing and controlling viral illnesses in fish populations, the vaccination approach may help decrease the use of antibiotics in fish populations (*Hoelzer et al., 2018*; *Ma et al., 2019*). This is because the goal of vaccination is to stimulate the immune system so that it can form a long-lasting immunological memory and a stronger immune response when exposed to the pathogen during infections (*Palgen et al., 2021*). Therefore, the close relationship between immune system stimulation and the discovery of epitopes was demonstrated, which is a considerably interesting aspect when formulating vaccines to create efficacious epitope vaccines (*Palatnik-de Sousa, Soares I da & Rosa, 2018*). The design of vaccines based on epitopes required the antigenic peptides visible on the antigen-presenting cell (APC) and target cell surfaces to be identified (*Dudek et al., 2010*; *Mugunthan & Harish, 2021*). Antigens are any substances that induce immune systems to create antibodies to combat the issue and serve as sites of interactivity between the antibody, the B cells and T helper ($T_H$) cells, as well as the molecules of the antigen. Such a site of interaction is referred to as an epitope (*Marshall et al., 2018*). Antibodies recognise antigens *via* interaction at the molecular level between paratopes (that is, the residues of the antibody implicated when binding occurs) and the interacting regions (epitopes) of the targeted molecules (antigens) (*Jespersen et al., 2019*), as illustrated in Fig. 1.

B-cell epitopes (BCEs) and T-cell epitopes (TCEs) are the two types of epitopes (TCEs). The B-cell epitope is a portion of an antigen that is connected to the immunoglobulin or antibody. B-cells recognise BCEs, which comprise a solvent area exposed to an antigen. Toxins and pathogens are neutralised by B-cell receptors (BCR), which are secreted or generated on their surface to target them with great specificity (antibodies) and thereby identify them for destruction (*Sanchez-Trincado, Gomez-Perosanz & Reche, 2017*; *Bukhari et al., 2022*). In the mapping of B-cell epitopes, predictors based on structures are becoming more popular because of the growing number of antibody-antigen complexes in the PDB and IMGT/3Dstructure-DB whose structures are three-dimensional (3D), in

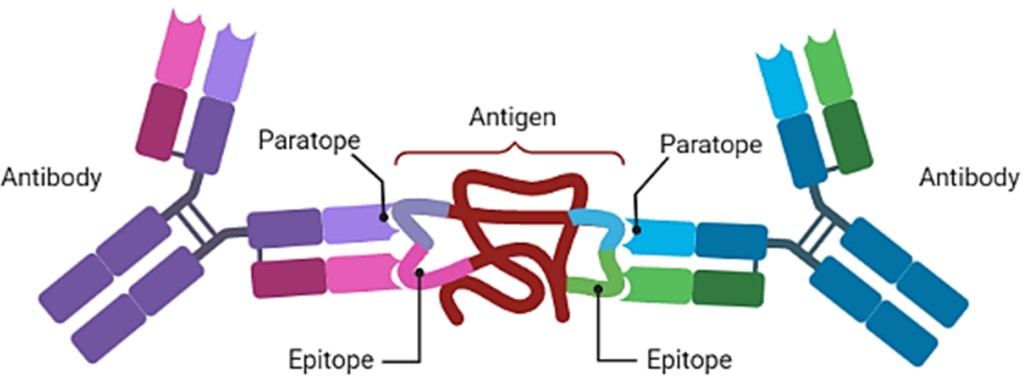

**Figure 1  An antibody with two paratopes.** These two paratopes are capable of binding to two pathogens. Non-covalent chemical interactions between epitopes and paratopes boost antigen–antibody binding. Created with BioRender.com.

addition to the capacity for continuous and discontinuous epitopes (also called linear and conformational epitopes, respectively) to be anticipated (*El-Manzalawy & Honavar, 2010*; *Soria-Guerra et al., 2015*; *Galanis et al., 2021*). To predict B cell epitopes, the majority of the current approaches employ antigen amino acid sequences such as ABCpred (*Malik et al., 2022*), IEDB B-cell epitope tools (*Vita et al., 2019*), SVMTriP(*Yao et al., 2012*), BCPred (*El-Manzalawy, Dobbs & Honavar, 2008*), LBtope (*Singh, Ansari & Raghava, 2013*), and BepiPred 2.0 (*Jespersen et al., 2017*). Meanwhile, the prediction of conformational B cell epitopes has involved various approaches, such as DiscoTope−2.0, BEpro (formerly known as PEPITO) (*Sweredoski & Baldi, 2008*), ElliPro (*Ponomarenko et al., 2008*), EPCES (*Liang et al., 2009*), EPSVR (*Liang et al., 2010*), EPMeta (*Liang et al., 2010*), Epitopia (*Rubinstein et al., 2009*) and SEPPA (*Sun et al., 2009*).

In contrast, a T-cell epitope is a peptide obtained *via* an antigen. They can be recognised by particular receptors named T-cell receptors (TCR) when they bind to key histocompatibility complex (MHC) molecules that appear on the surfaces of APC cells (*Sharma & Holt, 2014*; *Bukhari et al., 2022*). TCEs in complex with MHC proteins are recognised by two group subsets of T cells *i.e.,* T helper ($T_H$) or CD4$^+$ T cells and cytotoxic T lymphocytes (CTL) or CD8$^+$ T cells with different functionality (*Wieczorek et al., 2017*; *Marshall et al., 2018*). A $T_H$ cell response is produced when TCRs on CD4$^+$ T cells become bound to MHC class II–peptide complexes, which are frequently created in a professionally made APC. In contrast, CTL responses are elicited when TCRs on CD8$^+$ T cells can fix to MHC class I–peptide complexes that nucleated cells present (*Lundegaard, Lund & Nielsen, 2012*; *Sanchez-Trincado, Gomez-Perosanz & Reche, 2017*). Cytotoxic T lymphocytes (CTL) that have been activated produce cytokines, which cause them to divide and destroy the infected cells. Similarly, several of them transform into memory T cells (*Kar et al., 2020*) (Fig. 2B). Similarly, active cytokines cause B-cells to develop into plasma cells and memory B cells. Consequently, the activated plasma cell or B-cell releases antibodies or immunoglobulins (Igs) that are responsible for clearing the infection. Type of Igs also differ in the bony fish group, such as teleost fish (IgM, IgD, and IgZ/T), cartilaginous
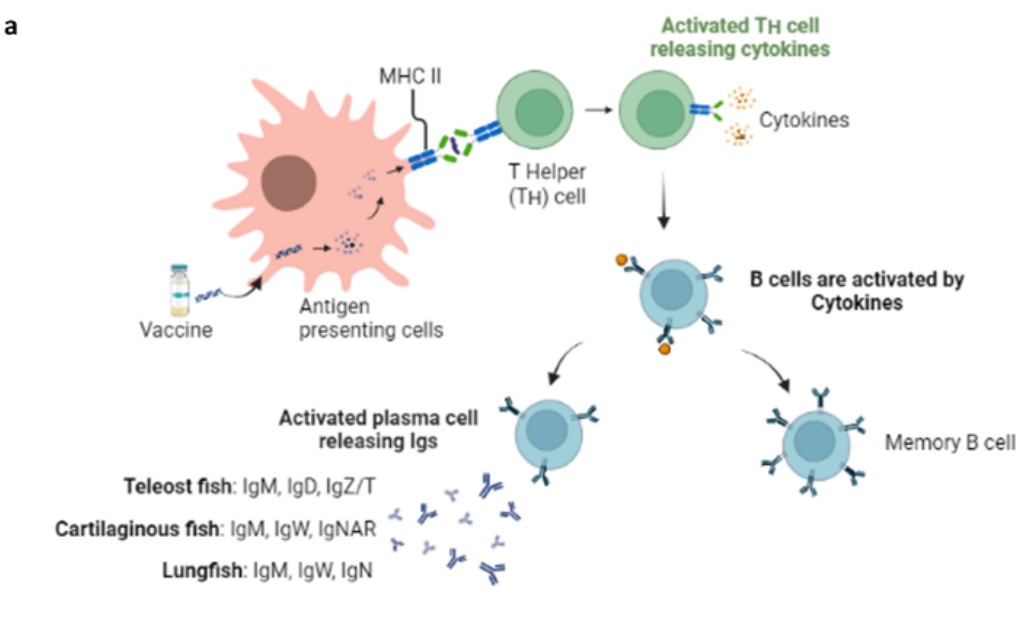

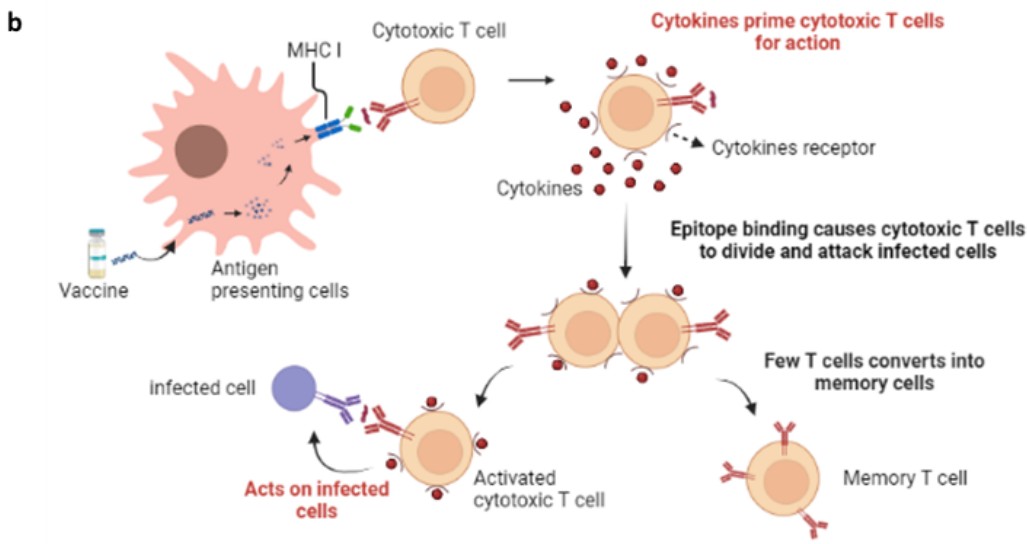

**Figure 2** **Immunological basis of the fish vaccine.** (A) Humoral immune response. (B) Cell-mediated immune response. Created with BioRender.com.

fish (IgM, IgW, IgNAR), and lungfish (IgM, IgW, IgN) (*Smith, Rise & Christian, 2019*) (Fig. 2A). This fact indicates that the strength of the MHC molecule's epitope binding is a key factor in determining T-cell epitope immunogenicity (*Mahendran et al., 2016a*; *Ogishi & Yotsuyanagi, 2019*).

The three crucial stages of immunogenicity in T-cell epitopes are as follows: antigens are processed, peptides attach to an MHC molecule, and cognate TCRs recognise this.

When determining a TCE, MHC-peptide binding utilises the greatest selectivity of the three steps. As a result, the primary basis for anticipating TCEs is peptide-MHC binding prediction (*Sanchez-Trincado, Gomez-Perosanz & Reche, 2017*; *Antunes et al., 2018*; *Feng, Zeng & Ma, 2021*). Numerous computational methods have been investigated and reviewed to predict TCEs and MHC-binding peptides, wherein both are computed based on binding matrices, binding motifs, decision trees, artificial neural networks (ANN), hidden Markov models (HMM), support vector machines (SVM), homology modelling and protein docking techniques as well as quantitative structure–activity relationship (QSAR) analysis (*Tong & Ren, 2009*; *Fleri et al., 2017*; *Kar et al., 2018*). HTL and CTL epitopes can be predicted using a variety of current bioinformatics tools, such as the IEDB database, RANKPEP server (*Reche et al., 2004*), ProPred (*Singh & Raghava, 2001*; *Reynisson et al., 2020*), NetMHCIIpan 3.2 (*Jensen et al., 2018*), NetCTL−1.2 (*Larsen et al., 2007*), ProPred1 (*Singh & Raghava, 2003*), NetMHCpan−4.1 (*Reynisson et al., 2020*), MHCpred 2.0 (*Guan et al., 2006*), EpiJen (*Doytchinova, Guan & Flower, 2006*), CTLPred and Expitope (*Haase et al., 2015*).

Although the majority of the B- and T-cell prediction tools have been developed and trained using data derived from human and mammalian major histocompatibility complex (MHC) or human leukocyte antigen (HLA) alleles, these tools are still relevant and applicable in vaccine design for aquaculture species. Despite the differences in MHC and HLA alleles between human and fish species, it is noteworthy that the similarities in the immune mechanisms involving B- and T-cell responses are crucial in the selection of antigens for vaccine design. The specific MHC for antigen presentation differ between human and fish species, but the conserved regions in antigens contribute to stimulating cross-reactive immune responses across species. The prediction tools are able to identify epitopes within these conserved regions, which could be recognized by fish immune cells and eventually lead to a specific immune response against the target pathogen. In addition, the functional similarities of immune cell receptors (B- and T-cell receptors) between fish and human enables the application of these tools to predict potential epitopes that are immunogenic in fish, based on the knowledge of B- and T-cell receptors in human. These tools have been utilized and demonstrated success in predicting epitopes for vaccine design against tilapia lake virus (*Islam et al., 2022a*) and *Edwardsiella ictaluri* in Nile tilapia (*Machimbirike et al., 2022a*), *Streptococcus iniae* (*Forouharmehr et al., 2022a*), *Flavobacterium columnare* (*Mahendran et al., 2016b*), and against *Ichthyophthirius multifiliis* (*Ghosh et al., 2023*). Nonetheless, experimental validation studies are indispensable. Candidate epitopes predicted by the immunoinformatics tools can be chemically synthesized and tested *in vitro* or *in vivo* to evaluate the immunogenicity and efficacy of immune activation in fish.

## Construction of multi-epitope vaccine
### Adjuvant selection
Peptide-based vaccinations, also known as epitope vaccines, are potential immunotherapeutic options and have been shown to have considerable advantages over conventional vaccines in multiple studies. However, when utilised alone in vaccine design, epitope

vaccines are linked with low protection, which can be overcome by conjugating antigenic epitopes with adjuvants and helper peptides. Several roles for vaccine adjuvants have been proposed; (i) dose-sparing strategy; (ii) accelerating seroconversion rates by enhancing antibody and cell-mediated immune responses; (iii) diversifying the adaptive immune profile and (iv) enhancing vaccine production by using smaller amounts of antigen (*Honda-Okubo, Baldwin & Petrovsky, 2021*; *Lemoine et al., 2021*). Toll-like receptor (TLR) agonists, antimicrobial peptides and helper peptides are the most common adjuvants used in peptide-based vaccine construction (*Shanmugam et al., 2012*; *Gupta et al., 2020*). TLRs also known as pattern recognition receptors, recognise common surface antigens found on microbes and act as a bridge between innate and adaptive immunity. In addition to TLR agonists, helper peptides and adjuvants derived from bacteria are used to boost the immune effects of peptide-based vaccines, including PADRE, Hsp70, $\beta$-defensin, bacterial toxins, cell wall components, flagellin, lipopolysaccharides (LPS), nucleic acids, and CpG oligodeoxynucleotides (ODN) (*Gries et al., 2019*; *Wang, Chen & Wang, 2019*; *Wangkahart, Secombes & Wang, 2019*; *Liang et al., 2020*).

Using bioinformatics in vaccine research and development has allowed for improved vaccination formulations and adjuvant selection. The development of adjuvants can be guided by databases including information on PRRs and their ligands. Numerous databases can be employed to select adjuvants, including:

1. Vaxjo (https://violinet.org/vaxjo/) is a web-based vaccine adjuvant database that includes approximately 400 vaccines that use an adjuvant against over 80 pathogens, cancers, or allergies (*Sayers et al., 2012*).

2. VaccineDA (Vaccine DNA adjuvants). This web-based resource (https://webs.iiitd.edu.in/raghava/vaccineda/) was developed to design immunomodulatory oligodeoxynucleotides (IMODN) -based vaccine adjuvants (*Nagpal et al., 2015*).

3. imRNA (https://webs.iiitd.edu.in/raghava/imrna/) is used to predict and design potential immunomodulatory RNA-based vaccine adjuvants (*Chaudhary et al., 2016*).

4. VaxinPAD (https://webs.iiitd.edu.in/raghava/vaxinpad/) employs SVM-based models to design peptide-based vaccines and allows users to perform virtual screenings that incorporate data from experimentally validated immunomodulatory peptides (*Nagpal et al., 2018*).

### Linker selection

Linkers, also known as 'spacers', are essential components in the design of multi-epitope vaccines (MEV) or peptide-based vaccines. They are critical for interdomain interactions, structural stability and functionality of vaccines. Fusion of epitopes without suitable linkers can result in negative outcomes such as 3D structural misfolding, low yield in vaccine production and bioactivity impairment. Despite their importance in recombinant MEV technology, the selection and rational design of linkers have not yet been thoroughly investigated. Flexible, rigid, and cleavable linkers are the three groups into which structural linkers can be categorised. To ensure flexibility of movement and interactivity between associated protein domains, those in the first linker group contain high levels of small and hydrophilic amino acids like glycine and serine. In contrast to flexible linkers, rigid linkers

may be utilised more effectively to maintain the desired stability or bioactivity between fusion protein domains. Multiple alanine and proline residues in these linkers exhibit a stiff structure that reduces interactions and separates the functional domains of the designed antigens (*Gräwe et al., 2020*). Cleavage linkers, on the other hand, are utilised to divide domains or peptides by proteolytic cleavage in order to decrease steric hindrance and achieve the independent biological function of a single domain when the linker is cleaved (*Leung et al., 2020*; *Poreba, 2020*)

Linker selection is influenced by the amino acid arrangement, length, hydrophobicity, secondary structure, and potential interaction with other immunogenic construct components. Linker DB which was created by Integrative Bioinformatics VU (IBIVU) at the Vrije Universiteit of Amsterdam, is the most recent database of linker peptides that enables the selection of prospective linkers for novel fusion proteins. This system provided a list of possible linkers based on the user-searched linker length, solvent accessibility, sequence motif, and protein source. Choosing which criteria to apply enables the user to opt for the linkers they require, depending on the conformation, adaptability, and stability needed to ensure the proteins function biologically in their natural environments.

### Prediction of vaccine antigenicity, allergenicity, toxicity and physicochemical properties

When designing and developing efficacious and safe candidates to use as vaccines, the vaccine constructs need to contain robust antigenicity but retain low toxic and allergenic levels. With advances in peptide synthesis, it is now feasible to fine-tune the physicochemical characteristics of peptides by including significant biochemical changes, maximising peptide functionality to reduce toxicity and allergenicity without limiting therapeutic effectiveness. A constructed vaccine sequence's physicochemical properties are often determined using the PortParam server (https://web.expasy.org/protparam/), which can compute the composition of amino acids, molecular weights (Mw), isoelectric points (pI), instability indices, predicted half-lives, and grand averages of hydropathicity (GRAVY). The isoelectric focusing approach can be employed to induce the isoelectric point (pI) computations to create buffer systems to purify proteins. Stability is predicted to apply to proteins when their instability index is below 40. A protein's aliphatic index (AI) refers to the relative volume taken up by its aliphatic side chain, which features, for instance, the amino acids alanine, valine, isoleucine, and leucine. Higher AI values will increase the thermal stability of globular proteins across a wider temperature range. A lower GRAVY value indicates a hydrophilicity pattern that is better suited for interaction with water.

The most significant requirement for efficient protein design is the antigenicity of the vaccine candidate, a high antigenicity score is expected to result in a greater immune response. A protein's antigenicity could be predicted by a range of servers, including Vaxijen (http://www.ddg-pharmfac.net/vaxijen/VaxiJen/VaxiJen.html) and ANTIGENpro (http://scratch.proteomics.ics.uci.edu/). The former method of predicting antigens is free from alignment and utilises the transformation of protein sequences through auto cross-covariance (ACC) to form uniformly developed vectors of the key attributes of amino acids, thus circumventing the limitations of methods that depend on alignment (*Flower*

*et al., 2017*). It was designed to allow antigen categorization exclusively based on protein physicochemical characteristics, without relying on sequence alignment. ANTIGENpro employs a pathogen-independent and sequence-based approach for predicting protein antigenicity (*Magnan et al., 2010*). This server predicts the entire protein antigenicity and the algorithm is trained to utilise reactivity data collected from protein microarray analysis for five pathogens including fungi, parasites, viruses, bacteria and tumours.

Toxicity is the ability of a material to cause damage in a live organism by destabilising and interfering with normal cellular activity. The ToxinPred tool (https://crdd.osdd.net/raghava/toxinpred/) can predict the toxicity of the computationally produced vaccine. ToxinPred is a support vector machine-based (SVM) approach to predicting peptide toxicity from sequence information using a position-specific scoring algorithm. ToxinPred was trained on a collection of known toxic and non-toxic peptides from the Universal Protein Resource (*Gupta et al., 2013*). The Toxins and Toxins Target Database (T3DB) which combines over 42,000 toxin data points with extensive toxin target information, is another resource that can be used to predict the toxicity of a vaccine candidate. It predicts if the constructed peptide vaccine may induce hypersensitivity responses. To determine the allergenicity of the potent vaccine, several services, such as AllerCatPro (*Maurer-Stroh et al., 2019*), AlgPred (*Sharma et al., 2021*), and AllerTop (*Dimitrov et al., 2014*) can be used to identify the allergenicity of the potent vaccine. In the former method, a protein's allergenic potential is predicted through its three-dimensional structure and the similarity of its sequence of amino acids to the data in a library of identified protein allergens. AlgPred allows the prediction of allergen using multiple allergenicity prediction approaches based on IgE epitope mapping, MAST motif alignment, allergen-representative peptides (ARPs) BLAST, support vector machines, and hybrid approaches. Finally, AllerTOP−2.0 employs a technique that is based on a protein's physicochemical similarity to known allergens.

### Structural modelling, assessment, and validation

A strategy based on structures does not rely exclusively on data from binding and information about sequences; instead, it leverages structural data and computation-based approaches created in structural biology so that binders of potential suitability can be identified. Vaccine sequences are received by a website that predicts protein structures so that three-dimensional structure models can be created once the incorporation of BCEs and TCEs has occurred by utilising suitable linkers and intramolecular adjuvants. The molecular structure of MHC molecules and their interactions with peptides can be utilised to create complex 3D models with other peptides, aiding in the explanation of atomistic aspects of molecular structures connected to biological system operation.

Emerging developments in machine learning and deep learning offer unprecedented opportunities for aquaculture, particularly in the development of effective fish vaccines. These innovations could be used to devise better vaccines for fish. AlphaFold uses deep learning models to predict the 3D structures of proteins based on their amino acid sequences. It has tremendous ramifications in fish health management and aquaculture that cannot be understated as we move from theoretical to practical applications. This instrument enables researchers to navigate the complexity of protein structures, which serve
as the blueprint for comprehending the antigen-antibody interaction, the linchpin of the immune response. Alphafold can predict the structure of several outer membrane proteins (OMPs), such as monomeric outer membrane protein A (OmpA), outer membrane protein 34 (Omp34), and a nucleoside-specific outer membrane transporter protein Tsx (OmpTsx) from rainbow trout (*Oncorhynchus mykiss*) afflicted with *Acinetobacter johnsonii* (*Bi et al., 2023*). This demonstrates the importance of AlphaFold in the design of subunit vaccines, which could exploit structural similarities in proteins across pathogens to enhance pathogen resistance in aquaculture environments. AlphaFold emerges as a potent instrument that revolutionises the landscape of vaccine development in aquaculture. With accurate protein structure prediction, we can accelerate the identification of potential vaccine targets and the development of effective treatments for diseases that threaten aquaculture production. With a reliable structural model, researchers can design vaccines that expose these epitopes to the fish immune system in a highly specific manner, thereby increasing the probability of a robust immune response.

Similarly, RosettaFold uses machine learning techniques to predict protein structures and could play a comparable role in the development of aquaculture vaccines. The precise prediction of the structures of proteins associated with fish diseases could assist in the development of multi-epitope vaccines. Such vaccines would target multiple proteins or multiple parts of a pathogen's protein, evoking a more robust immune response. This is especially important when considering the enormous variety of pathogens that can affect fish and the inherent difficulty of designing vaccines that provide protection for a variety of fish species. Utilising these sophisticated predictive tools could usher in a new era of fish vaccine development, resulting in not only more effective but also more cost-effective vaccines. This would considerably improve disease resistance and sustainability within the aquaculture industry, which is essential given the increasing reliance on aquaculture for food production worldwide. However, despite the promising potential of these tools, it is essential to note that their outputs are computational predictions. Consequently, any vaccines developed based on these predictions must undergo rigorous laboratory testing and clinical trials to ensure their safety and effectiveness. In addition, ongoing research and refinement of these computational tools will be necessary to improve their predictive accuracy, thereby maximising their utility in furthering aquaculture vaccine design.

Predicting protein tertiary structure can be accomplished using one of three methods: (1) Homology modelling, (2) threading, and (3) *ab initio* prediction. This method involves searching the Protein Data Bank (PDB) for structure-based similarities in the sequencing of the anticipated domains, from which a set of matches is generated according to E-values, alignment lengths, identities, and total scores. The basis for comparative modelling, which can be referred to as homology modelling, is the connection between target sequences and no fewer than one recognised three-dimensional structure belonging to the same family. Proteins that are aligned and have a higher percentage of identical residues imply evolutionary relationships. This method consists of several steps: (1) template identification and initial alignment, (2) alignment correction (3) backbone generation (4) loop and side-chain modelling, and (5) structure refinement and model evaluation (Fig. 3). Homology modelling is considered the most reliable method for predicting a protein's structure.

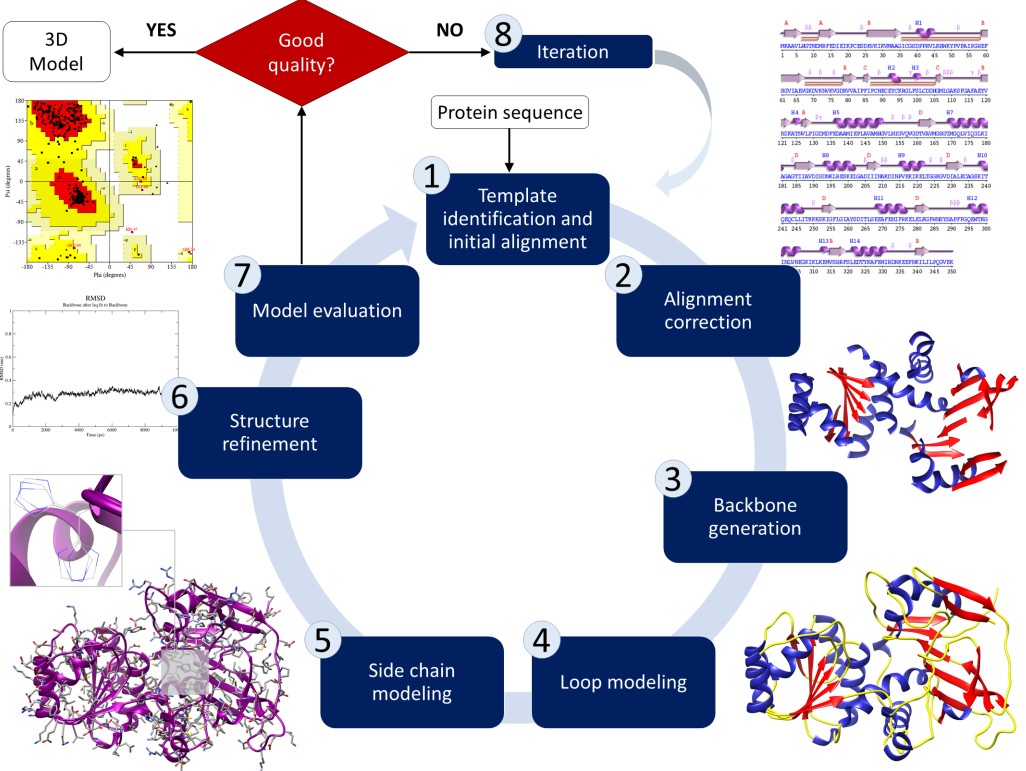

**Figure 3  Schematic illustration of the basic process of comparative modelling for protein structure prediction.** This method consists of several steps including template identification, initial alignment, alignment correction, backbone generation, loop and side-chain modelling, structure refinement and model evaluation.

Nevertheless, it can be complicated to identify suitable template structures that have high coverage of sequences and sequence identities. In general, template structures with a coverage of less than 35% are considered unreliable templates. EasyModeller 4.0 (*Kuntal, Aparoy & Reddanna, 2010*), SWISS-MODEL (*Waterhouse et al., 2018*), Rosetta (*Leman et al., 2020*), and Phyre2 (*Kelley et al., 2015*) are the most often used internet servers for homology modelling.

Threading, also known as fold recognition, is an alternative method if homology modelling cannot be applied. By comparing a template sequence to a collection of structural folds, this approach returns a list of scores. A known peptide–MHC complex structure is utilised to predict the binding structures of other peptides to the same MHC molecule by maximising the alignment of the amino acid sequence and their 3D structural patterns. I-TASSER (*MacCarthy et al., 2022*), Phyre2 (*Kelley et al., 2015*), and RaptorX (*Wang et al., 2016*) servers are used for threading modelling. *Ab initio* protein modelling predicts 3D structures based on novel folds and can be utilised if the structure of interest is unavailable or if the sequence identity between the template and the protein of interest is less than 30%. Based on physical principles, this method involves computing all energy parameters of protein folding and determining the state with the lowest free energy. ROBETTA (*Park*

*et al., 2018*), TrRosetta (*Du et al., 2021*) and I-TASSER (*MacCarthy et al., 2022*) are servers that can be used to predict the *ab initio* protein structure.

Structural validation is an important step in protein modelling to assure the quality of the models. Energy minimisation and structural refinement are applied to each model to enhance the three-dimensional structures' quality so that structural errors and steric conflicts in the structure of each protein can be eliminated. Several structure validation tools (*Razali et al., 2016*; *Razali & Shamsir, 2020*) such as PROCHECK (http://www.ebi.ac.uk/thornton-srv/software/PROCHECK/), Verify3D and ERRAT (https://saves.mbi.ucla.edu/) can be used to assess model quality before and after refining. PROCHECK provides a Ramachandran plot that calculates phi–psi torsion angles for each residue and analyses the overall stereochemical quality of 3D structures of protein models. Verify3D evaluate the compatibility of an atomic model (3D) with its amino acid sequence (1D) while the ERRAT server assesses the overall quality factor for nonbonded atomic interactions. The model quality assessment is critical for determining the overall correctness of the structure as well as the local accuracy of each protein fragment. Selecting the best protein modelling approach is dependent on the availability of known homologues, the folds of known structures, and the quality of 3D structures. Figure 4 depicts a schematic illustration of protein structure decision-making prediction based on several modelling methodologies.

In addition to the advancements made in protein structure prediction, the development of self-assembling immunogens is also making headway in the vaccine research landscape.

Self-assembling immunogens, such as virus-like particles (VLPs) and self-assembling protein nanoparticles (SAPNs), are used in the development of fish vaccines to enhance their immunogenicity and stability (*Ma et al., 2019*; *Abudula et al., 2020*; *Nakahira et al., 2021*). These immunogens are formed from the self-assembly of viral or bacterial proteins, which mimic the shape and size of native virions. Self-assembling peptides can also act as adjuvants themselves by forming an antigen depot, directing vaccines to antigen-presenting cells (APCs), and enhancing immune-cell priming (*Abudula et al., 2020*). These self-assembling structures, often designed as nanoparticles, mimic the native size and shape of viruses or bacteria and present multiple copies of an antigen or epitope on their surface. This form of antigen presentation stimulates a robust immune response by imitating the repetitive antigenic patterns of many pathogens. The use of self-assembling immunogens in fish vaccines offers several advantages, including precise antigen display, enhanced immunogenicity, and stability (*Rudra et al., 2010*).

Within the field of aquaculture, the potential benefits of self-assembling immunogens are multifaceted. Primarily, these structures could significantly enhance the immune response in fish. Due to the repetitive, ordered array of antigens that these nanoparticle vaccines present, they can stimulate the immune system more effectively, leading to a more potent and enduring response than traditional vaccines. Moreover, the versatility of self-assembling immunogens opens doors to designing broad-spectrum or multivalent vaccines. Such vaccines could potentially combat multiple strains or species of pathogens, addressing the substantial challenge of pathogenic diversity in aquaculture. On a practical note, self-assembling immunogens might present advantages in terms of stability and

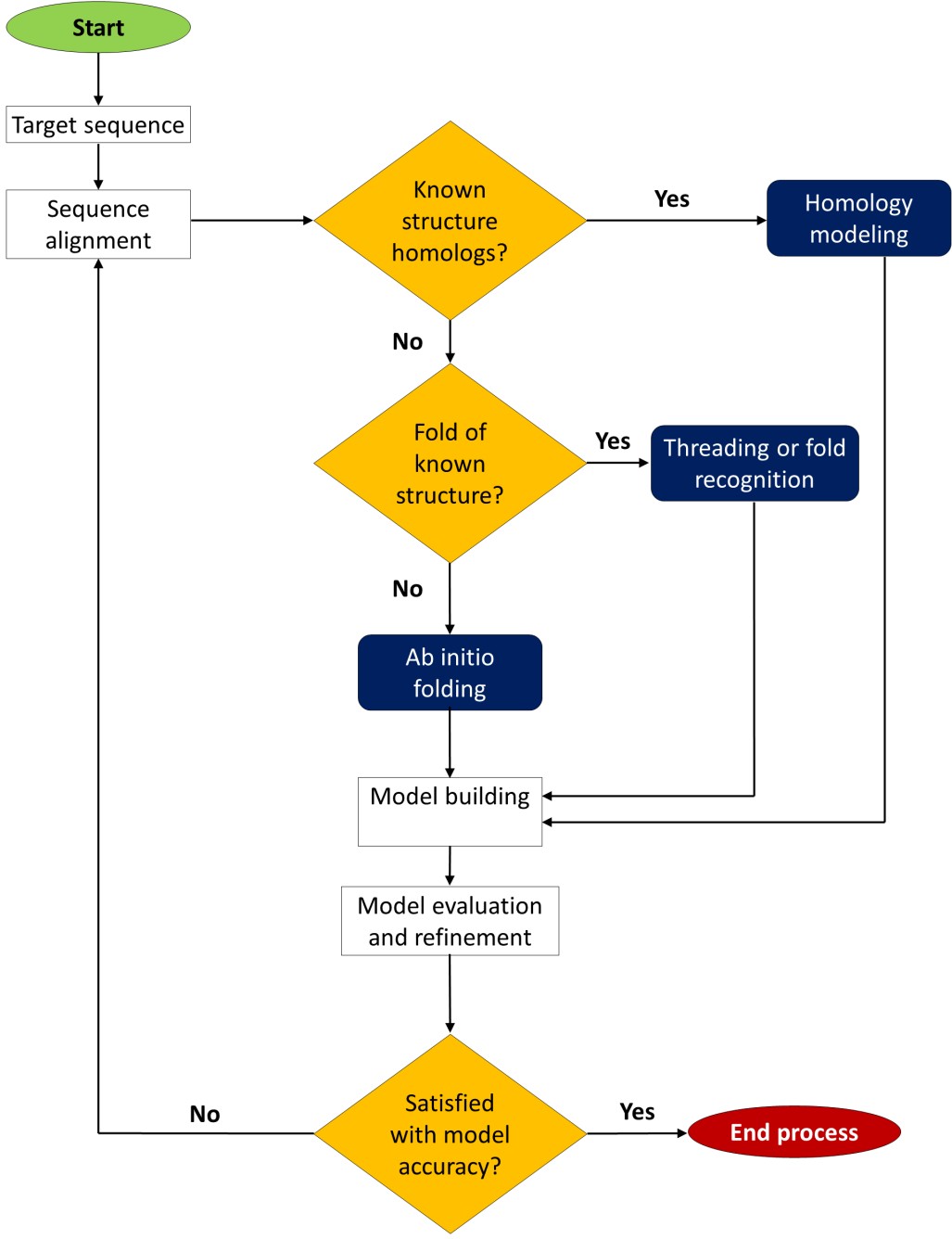

**Figure 4 Decision-making chart for protein structure prediction method.** The prediction of the 3D structure of a protein can be carried out with one of these three approaches: homology modelling, threading, or *ab initio* prediction.

scalability. Assuming the initial design and production processes are fine-tuned, these vaccines could potentially be produced on a large scale and might exhibit enhanced stability under a variety of environmental conditions, an essential consideration for aquaculture
operations worldwide. However, the development of self-assembling immunogens for fish vaccines is not without challenges. The design and production processes for these vaccines can be complex, necessitating further research to streamline them. Furthermore, experimental validation of the safety and efficacy of these vaccines in target fish species remains paramount. In this context, computational methods could play a significant role.

Self-assembling immunogens can be developed using computational methods such as AlphaFold and RosettaFold (*Castro et al., 2022*; *Morales-Hernández, Ugidos-Damboriena & López-Sagaseta, 2022*; *Olshefsky et al., 2022*). These computational methods enable the design of self-assembling immunogens by predicting the structure of proteins and their interactions with other proteins. AlphaFold uses deep neural networks to predict protein structures with high accuracy, while RosettaFold uses a combination of computational methods to predict protein structures. These computational methods can be used to design self-assembling immunogens with specific epitopes or antigens, which can enhance their immunogenicity and specificity.

The use of computational methods in the development of self-assembling immunogens offers several advantages, including the ability to design immunogens with specific properties and the ability to optimise immunogenicity and stability (*Castro et al., 2022*; *Morales-Hernández, Ugidos-Damboriena & López-Sagaseta, 2022*; *Olshefsky et al., 2022*). These tools could help design self-assembling immunogens by predicting the structures of antigens and then guiding the design of self-assembling proteins or peptides that best present these antigens. Overall, self-assembling immunogens provide a promising avenue for fish vaccine development. Their potential, coupled with the burgeoning capabilities of machine learning and deep learning tools, underscores an exciting frontier in aquaculture vaccine design. Nevertheless, it is crucial to maintain a balance between this optimism and a realistic understanding of the extensive validation and optimisation these novel approaches require.

## Molecular interaction of immunogenic vaccine
### Molecular docking and molecular dynamics simulation
Protein-peptide docking is another important tool for predicting the efficacy of a vaccine. Unlike the costly and lengthy approaches involved in crystallising and structurally resolving a TCR-MHC complex, the computational tool of molecular docking can efficiently and cost-effectively enable intermolecular-level interactions within ligand–receptor complexes to be studied. This prediction is made using a software that includes (1) regeneration of all possible ligand structure formations, (2) placement of all ligand formations in a cavity of the active target protein position, and (3) scoring function based on free energy or binding energy. For molecular docking analysis, numerous software and tools have been created, including standalone applications such as Autodock Vina (*Eberhardt et al., 2021*), Autodock 4 (*Santos-Martins et al., 2021*), ZDOCK (*Vreven et al., 2020*), Glide (*Alogheli et al., 2017*) and GOLD (*Martin et al., 2020*). Several online servers, such as RosettaDock (*Marze et al., 2018*), ClusPro (*Alekseenko et al., 2020*), and HADDOCK (*Roel-Touris et al., 2019*) are also available to study protein-protein docking interactions. However, these web servers are not suitable for large-scale studies as they are limited to a single protein-protein

PeerJ ————————————————————————————

docking simulation. EpiDOCK (*Atanasova et al., 2013*) is also one of the structure-based servers used for MHC binding peptide prediction using dock score-based QM.

Molecular dynamics (MD) simulation is a method for studying the movement of molecules and atoms in a realistic molecular system by utilising a force field to model intramolecular and intermolecular interactions at the atomic level. Using this technique, which numerically solves the time-dependent behaviour of a molecular system on a microscopic scale, the structure and conformational changes of proteins, as well as their thermodynamic properties, are examined in depth. It can also be used to investigate the dynamics and binding mode of novel peptide vaccines, interactions between peptide vaccines and the receptor binding groove, residue specificity and dissociation of MHC peptide-protein complexes, and interactions between the T-cell receptor and the MHC–peptide complex. AMBER (*Pang, 2016*), GROMACS (*Kohnke, Kutzner & Grubmüller, 2020*), CHARMM (*Kim et al., 2020*), and NAMD (*Phillips et al., 2020*) are some of the most used force fields and MD simulation programmes for calculating binding free energies.

The creation of these computational methods will facilitate the molecular analysis of peptide vaccines and receptor interactions, thereby facilitating the design and development of possible vaccinations against fish diseases. To design a multi-epitope subunit vaccine targeting the fish pathogen, the following immunoinformatics steps will be sequentially applied: (1) screening of the fish pathogen proteome, (2) B- and T-cell epitope prediction, (3) construction of vaccine by joining together the epitopes, linkers, and adjuvants, (4) vaccine properties prediction, (5) vaccine 3D structure modelling, (6) molecular docking with TLRs, and (7) MD simulations for stability (Fig. 5).

## Immune mechanism and immunoinformatics in fish disease
### Immune mechanism prediction: TLR signalling pathways

To recognise an infection in an innate immune system, the toll-like receptor (TLR) is the receptor that has been researched extensively (*Palti, 2011*; *Li et al., 2017*). These receptors can also be referred to as a pattern recognition receptor (PRR) family that recognises, firstly, an external pathogen-associated molecular pattern (PAMP) derived from several types of microbial pathogen (*Zhang & Liang, 2016*) and, secondly, an internal damage-associated molecular pattern (DAMP) created by cells near death or tissues that have suffered damage (*Yu & Feng, 2018*). Different TLRs play an essential role in bridging the gap between innate and adaptive immunity by determining characteristics such as accurate identification and immune response to hazardous stimuli (*El-Zayat, Sibaii & Mannaa, 2019*).

Fish TLRs and the components involved in their signalling cascade share significant structural similarities with the mammalian TLR system. Despite this, the fish TLRs exhibit unique features and a wide range of diversity, which is likely due to their diverse evolutionary history and habitat. To date, thirteen TLR members have been discovered in mammals and each of which functions as a sensor for different PAMPs (*Palti, 2011*; *Wang et al., 2021*). In addition, TLRs are also present in fish where more than 21 TLRs have been reported so far (*Liao et al., 2017*). Fish have been shown to not possess TLR6, TLR10, TLR11, and TLR12. Furthermore, TLR4 which is absent in many species is found in some cyprinid fish,

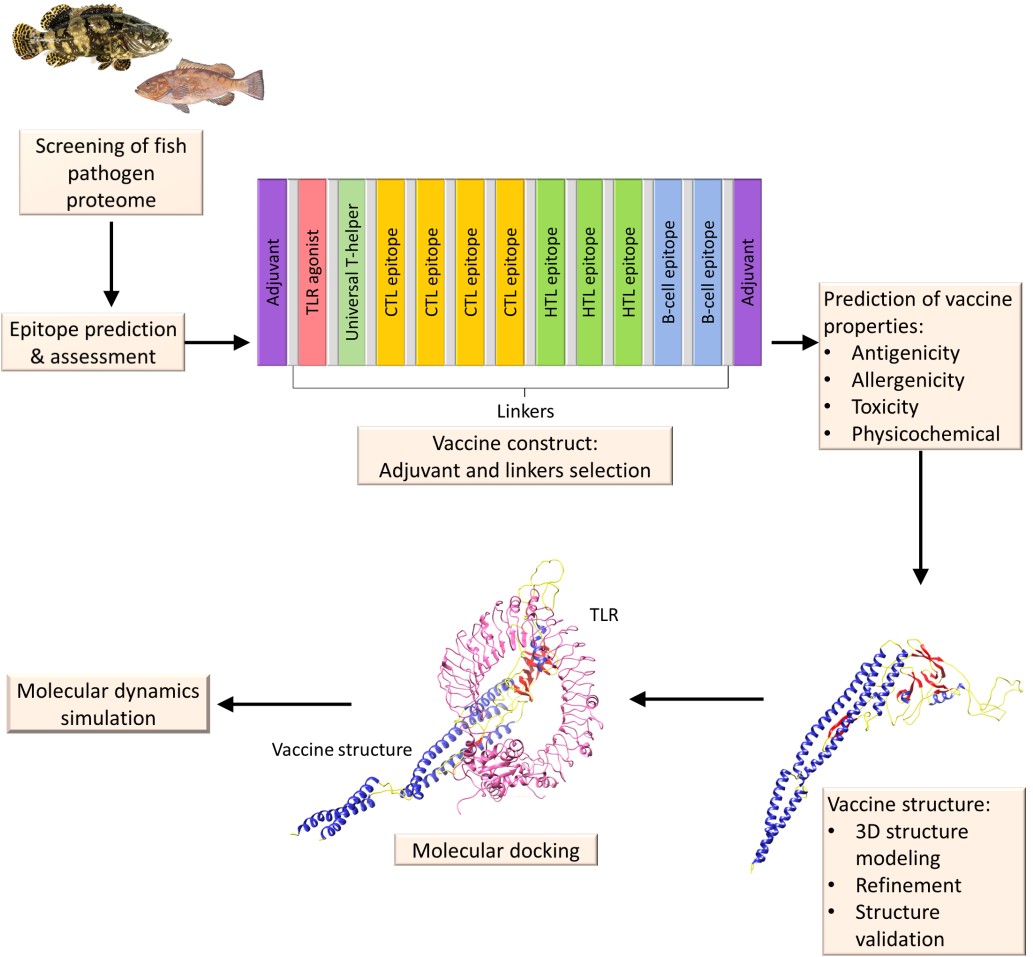

**Figure 5** **A diagrammatic description of the procedures involved in the *in silico* design of a multi-epitope vaccine for fish illnesses.** Beginning with proteome retrieval and continuing through multi-epitope vaccine design and its validation by molecular docking and MD simulation.

along with TLR5S, TLR18-TLR20, TLR23, and TLR25-TLR28, which are considered to be "fish-specific" TLRs (*Rebl, Goldammer & Seyfert, 2010*; *Palti, 2011*; *Wang et al., 2015*; *He et al., 2019a*).

Recent studies on fish TLRs have concentrated on identifying individual TLR members in diverse teleost fish species such as TLR7 and TLR8 in Barbel chub (*Squaliobarbus curriculus*) (*Jin et al., 2018*), TLR21, TLR22, and TLR25 in Dabry's sturgeon (*Acipenser dabryanus*) (*Qi et al., 2018*), TLR1-TLR3, TLR5, TLR7-TLR9. TLR13, TLR22, TLR25, and TLR26 in Walking catfish (*Clarias batrachus*) (*Priyam et al., 2020*). Furthermore, researchers have also examined what occurs when a pathogenic bacterium, virus, or ligand is used as a stimulator, with regard to expression profiles and signalling cascade genes (*Jiang et al., 2020*; *Muduli et al., 2021*; *Wang et al., 2021*). TLR ligands remain substantially unknown, especially in cartilaginous fish and lobe-finned fish (*Nie et al., 2018*; *Smith, Rise & Christian, 2019*). Several investigations detected TLR2, TLR3, TLR6, and TLR9 in the

transcriptome data of the grey bamboo shark (*Chiloscyllium griseum*), (*Anandhakumar et al., 2012*; *Krishnaswamy et al., 2014*) while TLR3 was identified in the immunological response of the Nigerian spotted lungfish (*Protopterus dolloi*) (*Tacchi, Misra & Salinas, 2013*). Among all these TLRs, TLR4, TLR5, TLR9, and TLR14 are regarded as sensors of bacterial ligands while TLR3, TLR7, TLR8, and TLR22 are presumed viral ligands sensors (*Rebl, Goldammer & Seyfert, 2010*; *Rauta et al., 2014*; *Zhang et al., 2014*; *Wang et al., 2021*). Thus, in this review, the ligand specificity like PAMPs and signal pathways of fish TLRs are summarised in Fig. 6.

Theoretically, TLRs are activated when they recognise ligands which prompts the recruitment of adaptor molecules in the cytoplasm and the initiation of signalling cascades (*Kenny & O'Neill, 2008*; *Luo et al., 2020*). Both adaptors MyD88 (myeloid differentiation primary response 88) and TRIF (TIR-domain-containing adapter-inducing interferon-$\beta$) also known as TICAM-1 (TIR-containing adaptor molecule-1) are signals to downstream signalling pathways whereas the other one, TIRAP is primarily an adaptor for TLRs to connect to TRIF and MyD88, respectively (*Troutman, Bazan & Pasare, 2012*; *DeFranco, 2016*; *Farooq et al., 2021*) (Fig. 6). The activation of TLR signalling implicates at least two different pathways *i.e., via* the MyD88-dependent pathway which leads to the induction of various cytokines (IL-6, IL-8, IL-12, and TNF$\alpha$) and MyD88-independent pathway which is associated with the induction of IFN and maturation of dendritic cells (*Rauta et al., 2014*; *Farooq et al., 2021*).

Through the MyD88-dependent pathway, MyD88 utilizes its death domain to interact with IRAK4 (IL-1 receptor-associated protein kinase 4) to form the MyD88-IRAK4 complex. This complex then phosphorylates IRAK2 or IRAK1 and recruits TRAF6 (tumour necrosis factor receptor-associated factor 6) *via* ubiquitination (*Cao, Henzel & Gao, 1996*; *Fitzgerald & Kagan, 2020*). Studies have reported that IRAK2 has been lost or not identified in fish (*Zhang et al., 2014*; *Rebl et al., 2019*), however, another study has stated that IRAK2 was found in the West Indian Ocean coelacanth (*Latimeria chalumnae*) genome (*Li et al., 2018*). Following ubiquitination, TRAF6 interacts and activates the TAB1/TAK1/TAB2 complex whereby TAB1 activates TAK1 (transforming growth factor-$\beta$-activated kinase 1) while TAB2 serves as an adaptor that connects TAK1 to TRAF6. TAK1 is then coupled to the IKK complex which leads to IKKßphosphorylation and the subsequent translocation of the NF-$\kappa$ßcomplex into the nucleus. NF-$\kappa$ßcomplex containing p50 and p65 combines with gene transcription to induce proinflammatory cytokines such as IL-6, IL-8, IL-12, and TNF$\alpha$ (*Rebl, Goldammer & Seyfert, 2010*; *Rauta et al., 2014*; *Farooq et al., 2021*).

TAK1 simultaneously phosphorylates MAPKs (mitogen-activated protein kinases) and induces the activation of AP-1 (activating protein-1) (*Xu & Lei, 2021*). For example, MaTLR14, a fish-specific TLR14 was identified in an Asian swamp eel (*Monopterus albus*) which increased TRAF6 expression and phosphorylation of ERK (extracellular signal-regulated kinase) and p65, thereby activating the NF-$\kappa$ßcomplex and AP-1. As a result, this phenomenon stimulated the production of proinflammatory cytokines such as IL-6 and TNF$\alpha$ (*Zhang et al., 2014*; *Liu et al., 2022*). Additionally, TLR14 was identified in the majority of fish orders, including pufferfish, zebrafish, flounder, golden pompano, and lamprey (*Rebl, Goldammer & Seyfert, 2010*; *Wu et al., 2019*; *Sousa et al., 2022*), and

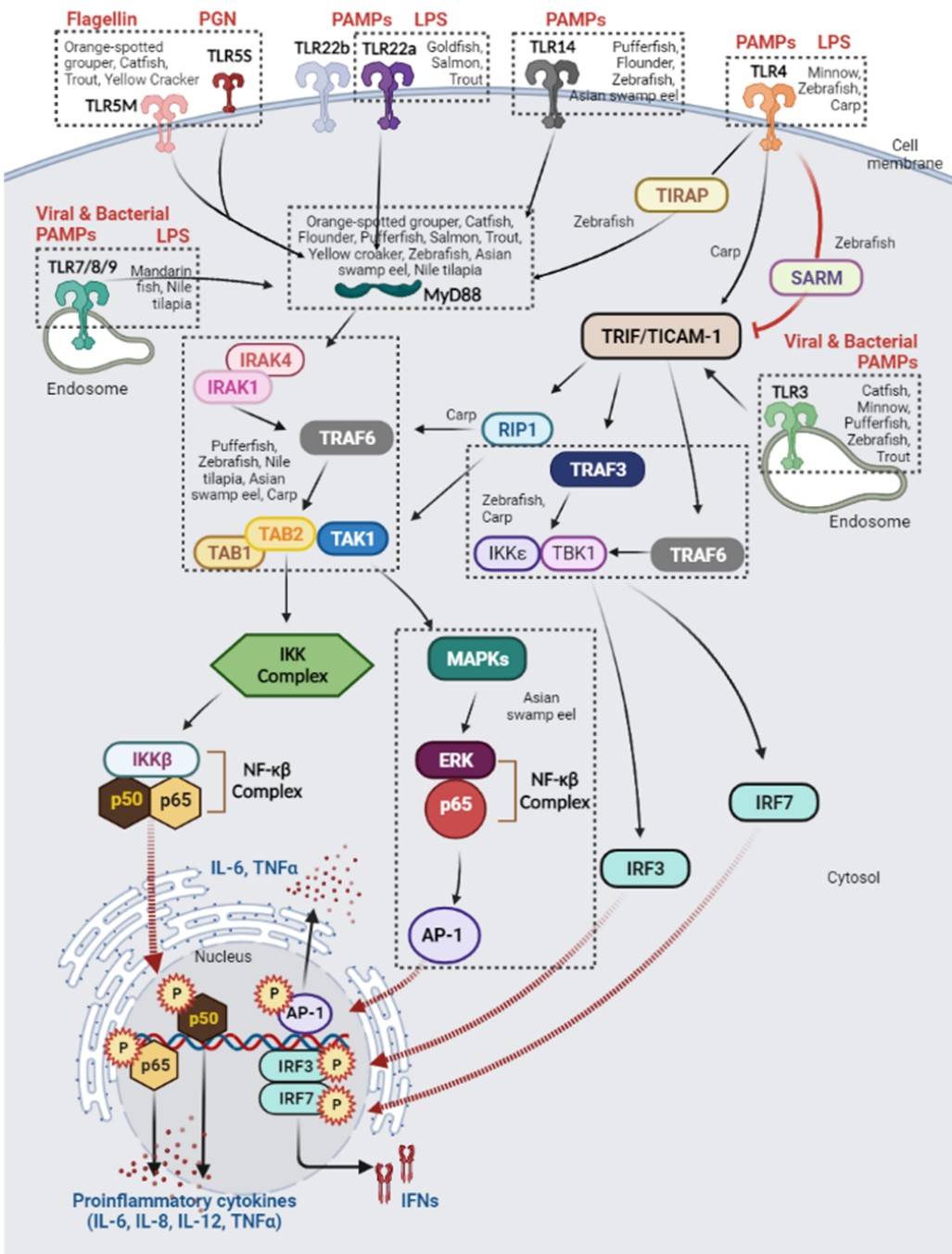

**Figure 6  Schematic illustration of immune mechanisms activation in different fish species (boxed by dotted lines) through Toll-like receptor (TLR) signalling pathways.** Modified from *Rauta et al. (2014)*, *Rebl, Goldammer & Seyfert (2010)*, and *Zhang et al. (2014)*. Created with BioRender.com.

interestingly, it shared similar features to TLR6 and TLR10 of mammals, despite the absence of both TLRs in fish signalling cascades (*Rauta et al., 2014*; *Liao et al., 2017*).

Additionally, nearly every poikilothermic vertebrate (for instance, an amphibian or a fish) has exhibited TLR22, although it is not present in mammals. TLR22 has a vital part to the role in activating adaptive immunity and initiating innate immunity. In addition, the MyD88 adapter is required to initiate the signalling cascade (*Paria et al., 2018*; *Ji et al., 2020*; *Gao et al., 2021a*). Similarly, to TLR4, TLR5 activates the MyD88-dependent pathway and consists of two distinct forms, the membrane TLR5 (TLR5M) and soluble TLR5 (TLR5S). Together, they detected the bacterial flagellin, despite having opposing roles in stimulating the signalling cascade. The flagellin induced the basal activation of NF-$\kappa$ß*via* TLR5M, causing the activation of TLR5S expression in the liver. TLR5S is efficient in binding to circulating flagellin and transporting the latter to the membrane TLR5 factor, which amplifies the signalling of danger in positive loop feedback pathways (*Rebl, Goldammer & Seyfert, 2010*; *Zhang et al., 2014*; *Jiang et al., 2017*; *Jiang et al., 2020*; *He et al., 2019b*). This phenomenon is observed in the stimulation of *Vibrio parahaemolyticus* flagellin in orange-spotted grouper (*Epinephelus coioides*) (*Bai et al., 2017*; *He et al., 2019a*) and large yellow croaker (*Larimichthys crocea*) (*Jiang et al., 2020*) as well as *Yersinia ruckeri* stimulation in rainbow trout (*Oncorhynchus mykiss*) (*Wangkahart, Secombes & Wang, 2019*) and channel catfish (*Ictalurus punctatus*) (*Jiang et al., 2017*).

In contrast to the three TLR family members TLR7, TLR8, and TLR9, bacterial and viral PAMPs do not induce their cell signalling cascades in endosomes after being activated by lipopolysaccharides (LPS). The MyD88-dependent pathway considerably elevated their expression levels in all investigated tissues of Nile tilapia (*Oreochromis niloticus*) and mandarin fish (*Siniperca chuatsi*) (*Gao et al., 2021b*; *Wang et al., 2021*). In addition, the foregoing results indicate that the MyD88-dependent pathway in fish is comparable to all TLRs except TLR3 (*Rauta et al., 2014*; *Zhang et al., 2014*; *Fitzgerald & Kagan, 2020*).

In addition, TLR4 in zebrafish (Danio rerio) uses alternative adaptor proteins such as TIRAP to recruit MyD88 to activate IRAK following the induction of LPS and PAMPs, as depicted in Fig. 6 (*Rauta et al., 2014*; *Li et al., 2017*; *Loes et al., 2021*). In addition to TIRAP, additional adaptor molecules, such as SARM (sterile alpha and HEAT/Armadillo motif-containing protein), have been identified in zebrafish. SARM is the only adaptor protein that inhibits TLR signalling by interacting with TRIF (red arrow in Fig. 6). Its expression inhibited the function of TRIF *via* the TLR3 and TLR4 pathways, whereas its silencing had the opposite effect (*Peng et al., 2010*; *Kanwal et al., 2014*; *Loring & Thompson, 2020*; *Luo et al., 2020*).

The recruitment of TRIF to TLR4 and TLR3 occurs, thereby promoting an alternative avenue which results in IRF3, NF-ß, AP-1, MAPKs, and IRF3 (interferon regulatory factor 3) being activated to produce proinflammatory cytokines and/or IFN1 (type I interferon) (*Kawasaki & Kawai, 2014*; *Hu et al., 2015*; *Nie et al., 2018*). In addition, NF-$\kappa$ßin the TRIF-dependent pathway can be activated *via* the recruitment of RIP1 and TRAF6 *via* the C-terminal RHIM domain and the TRAF6 binding motif, respectively (*Zhang et al., 2020*; *Liu et al., 2021*). Intriguingly, the activation of NF-$\kappa$ßin carp TRIF was consistent with the findings in large yellow cracker, orange-spotted grouper, and zebrafish, indicating that fish TRIF in the NF-$\kappa$ß-mediating signalling cascade has a conserved function (*Zhang et al., 2020*; *Liu et al., 2021*; *Zou et al., 2021*).

Unlike TLR4, TLR3 interacts directly with TICAM-1 (also known as TRIF) in the MyD88-independent pathway. The TLR3-TICAM-1 signalling pathway is one of the most important immune responses to RNA virus infection (*Rebl, Goldammer & Seyfert, 2010*; *Nie et al., 2018*; *Geng et al., 2021*). As a result, it induces IFN1 production by activating IFN3 and IFN7 *via* an interaction with TRAF3 and TRAF6 (*Li et al., 2017*; *Geng et al., 2021*). As shown in Fig. 6. TLR3 has been identified in numerous fish species, such as channel catfish (*Ictalurus punctatus*), rare minnow (*Gobiocypris rarus*), pufferfish (*Takifugu rubripes*), zebrafish (*Danio rerio*), and rainbow trout (*Oncorhynchus mykiss*).

To conclude, the TLR cascade involves the recruitment of components whose functions resemble each other or are identical in every species of vertebrate apart from fish, whose attributes are unique. More downstream components of this signalling cascade must be studied in bony fish (teleost fish, cartilaginous fish and lungfish) to elucidate the functional similarities and divergences of TLR signalling in a fish with bones or a mammal. The importance of this is due to TLR working to connect innate and adaptive immunity, which should facilitate the understanding of how a fish vaccine functions.

### Multi-epitope vaccine and treatment

Compared to conventional vaccinology, epitope-based chimeric (subunit) vaccines using an immunoinformatics approach offer many advantages such as not requiring microbial culturing, being less expensive to develop, taking less time to produce, outperforming numerous wet-lab experiments, and being specific and stable because they do not contain the entire organism (*Kar et al., 2020*; *Naz et al., 2020*; *Bukhari et al., 2022*). Due to the occurrence of MHC variants, an epitope-based vaccine targeting limited MHC alleles typically does not have the desired or equivalent effect on the fish population. Consequently, very promiscuous epitopes can simultaneously bind different alleles, allowing for the immunological response sought in a diverse fish population (*Wegner, 2008*; *Patronov & Doytchinova, 2013*; *Radwan et al., 2020*; *Šimková et al., 2021*). This approach, termed a multi-epitope vaccine, consists of a set of peptides that overlap, and it induces an immune response according to a short immunogenic sequence (*Zhang et al., 2012*).

A multi-epitope vaccine also uses certain design principles, such as TH, B-cell, and CTL epitopes, which can cause strong cellular and humoral immunity at the same time; many MHC-restricted epitopes, which a TCR from a different subset of T-cells can recognize; and many epitopes from different forms of antigens, which increases the number of bacteria and viruses that can be targetedFurthermore, multi-epitope vaccines involve the introduction of an adjuvant-capable element able to boost immunogenicity and offer durable immune responses. Moreover, undesirable elements that might cause an abnormal immune response or a detrimental side effect can be eliminated (*Saadi, Karkhah & Nouri, 2017*; *Zhang, 2018*; *Sami et al., 2021*; *Sanches et al., 2021*).

Recently, an *in-silico* method was able to accurately predict epitopes and multi-epitopes with remarkable responsiveness against *Streptococcus agalactiae*, *Streptococcus iniae*, *Edwardsiella tarda*, and *Flavobacterium columnarie* individually (*Forouharmehr et al., 2022b*; *Islam, Mou & Sanjida, 2022*). Pathogenic bacteria such as *Streptococcus agalactiae* have caused streptococcus's disease in tilapia aquaculture (*Toranzo, Magariños &*

*Romalde, 2005*; *Su et al., 2016*; *Sirimanapong et al., 2018*). Tilapia species such as Nile tilapia, *Oreochromi* s *niloticus* Linn. that are infected with *Streptococcus agalactiae* typically exhibit various symptoms such as dark skin pigment, dermal haemorrhages, hyperaemic gills, eye lesions, erratic swimming, spinal curvature, and diffuse epithelial tissue proliferation (*Geng et al., 2012*; *Su et al., 2016*; *Yi et al., 2019*). In addition, the currently available vaccines have limits in protecting fish against catastrophic death when infected with different strains of *Streptococci* sp. (*Pereira et al., 2013*; *Mishra et al., 2018*; *Pumchan et al., 2020*). Utilizing immunoinformatics, a strategy to design a multi-epitope vaccine was implemented to address this problem. As a result, two of five antigenic proteins (45F2 and 42E2) were predicted as the best candidates for constructing a multi-epitope vaccine and were subsequently shown to successfully protect against streptococcus's disease in tilapia (*Pumchan et al., 2020*).

Similarly, the most recent study reported by *Forouharmehr et al. (2022b)* described the effectiveness of multi-epitope vaccinations that employ a number of immunogenic proteins. The prediction of the epitopes involved six immunogenic Streptococcus iniae proteins: GAPDH, MtsB, ENO, Sip11, FBA, and SCPI. In this context, the most suitable multi-epitope vaccine was constructed by foreseeing various epitopes, such as the B-cell, T-cell, and IFN $\gamma$ epitopes of the immunogenic proteins and interleukin-8 (IL-8). An analysis was also conducted of the vaccine's antigenicity, physicochemical attributes, and secondary and tertiary structural forms, as well as different aspects deemed vital in the vaccine's development. Additionally, this study revealed that the developed vaccine's IL-8 domain had the highest level of binding affinity when docking with its receptor, and this was adapted with success so that it could be expressed in *Escherichia coli*. As a result, a stable vaccine with an antigenicity score of 0.936 and a 45-kDa molecular weight has been developed. This multi-epitope vaccine looks to be an effective candidate for preventing *Streptococcus iniae* infections in fish.

Additional pathogenic bacteria such as *Edwardsiella tarda* and *Flavobacterium columnare* also cause Edwardsiellosis and Columnaris diseases in the majority of fish species resulting in a high mortality rate among distinct populations of fish of varying ages (*Sudheesh et al., 2012*; *Declercq et al., 2013*; *Hirai et al., 2015*; *Zhou et al., 2018*). The *E. tarda*-infected fish displayed symptoms such as abnormal swimming, spiral movement, and floating near the water's surface. This virulent intracellular pathogen poses serious threats, particularly in the farming of catfish, flounder, turbot, yellowtail, and tilapia species (*Park, Aoki & Jung, 2012*; *Mahendran et al., 2016a*; *Miniero Davies et al., 2018*). In the meantime, columnaris disease caused by *F. columnare* mostly affects fish species such as goldfish, channel catfish, eels, tilapia, carp, perch, and salmonids. This virulent bacterium is found in the individual gill filaments and causes yellowish-brown lesions on the gills, fin, and skin (*Arias et al., 2012*; *Zhu et al., 2012*; *Mahendran et al., 2016a*; *Mitiku, 2018*). The need for novel vaccines against edwardsiellosis and columnaris illnesses has increased as a result of these concerns. Various antibiotics, including colistin, rifampin, oxacillin, and penicillin, have been used to control edwardsiellosis (*Mahendran et al., 2016a*), whilst quinolones and tetracyclines are used against columnaris infection (*Mitiku, 2018*). In spite of this, the extensive use of antibiotics results in the evolution of various drug resistances and has caused the enormous
deaths of farmed and wild fishes owing to bacterial infection (*Kumar et al., 2012*; *Beck et al., 2015*; *Abd El-Tawab et al., 2020*; *Preena, Dharmaratnam & Swaminathan, 2022*).

Although vaccines for columnaris treatment are now available, there is a risk of reversion in certain cases of live attenuated vaccines. Similarly, the possibility of a monovalent vaccine to protect all the susceptible fish hosts from *Edwardsiella* sp. is impossible. This is because the bacterium possesses different host-based genotypes such as serological, genetic, and antigenic heterogenous (*Park, Aoki & Jung, 2012*; *Mahendran et al., 2016a*; *Buján, Toranzo & Magariños, 2018*; *Bothammal et al., 2021*). Therefore, this risk of reversion could be prevented by predicting the B-cell and T-cell epitopes in peptide sequences and then developing an effective multi-epitopes vaccination using an immunoinformatics technique. The docked structure of peptide-MHC I complexes has been successfully modelled using two and five CTL epitopes of outer membrane proteins (OMPs) from *E. tarda* and *F. columnare*, respectively, according to a prior study described by *Mahendran et al. (2016a)*. Their interactions were studied using immunoinformatics tools. In addition, infection by other bacterial strains of *Edwardsiella* sp. in fish has also been reported recently. A multi-epitope chimeric protein, EiCh is composed of eleven B-cell epitopes and seven MHC II epitopes that were successfully constructed and expressed in *E. coli* BL-21 (DE3). As a consequence, 49.32-kDa recombinant EiCh protein induced a potent antibody response against *E. ictaluri* in Nile tilapia and striped catfish. This finding indicates that the immunoinformatics strategy for vaccine formulation studied in this study is essential for treating *Edwardsiella* sp. infections in fish species (*Machimbirike et al., 2022b*).

In addition to bacterial illnesses, viral infections have a negative impact on aquaculture productivity. For instance, viral encephalopathy and retinopathy are caused by the nervous necrosis virus (NNV), leading to extensive death rates in commercially farmed species of fish that can exceed 100% (*Hazreen-Nita et al., 2019*; *Kaushik, 2020*; *Michel-Todó et al., 2020*). In this regard, an *in-silico* method was utilised to design an epitope-based vaccine to protect grouper and sea bass fish species from NNV infections. Six antigenic epitopes were selected from a pool of one thousand and conjugated with adjuvant and linker peptides. As a result, the model of an engineered epitope-based vaccine showed good binding to toll-like receptor-5 (TLR5), a crucial elicitor of the immune response. This prediction would be useful prior to cloning and purifying the NNV 248-specific protein (*Joshi et al., 2021*).

In addition to affecting marine and shellfish species, the marine birnavirus (MABV) outbreak has a significant economic impact on aquaculture production (*Mancheva et al., 2021*; *Fu et al., 2022*). In shorter periods in standard culture conditions, MABV is the most pathogenic virus to have resulted in complete mortality, hence the limitations on being able to prevent this virus (*Crane & Hyatt, 2011*; *Diggles, 2016*; *Chen et al., 2019*; *Islam, Mou & Sanjida, 2022*). Thus, an immuno-informatics method was employed to construct an epitope-based vaccine against MABV by recognising the most pathogenic and antigenic proteins of MABV; RNA-dependent RNA polymerase (RdRp), polyprotein (PP), and major capsid protein VP2 (MCPVP2) of MABV. For all the proteins, the leading three CTL epitopes with the most appropriate adjuvants and linkers to ensure non-allergenity, immunogenity, and better solubility were anticipated so that the multi-epitope birnavirus (MEBV) could be designed. Using E. coli K12 as a model, codon optimisation was

conducted to improve the translational efficiency of the vaccine design. The codon was ultimately modified, and *in silico* cloning using the E. coli K12 expression host, pET28a (+) vector, was effective. This potential peptide vaccine might be an effective MABV preventive strategy (*Islam, Mou & Sanjida, 2022*).

Using immunoinformatics methodologies prior to conducting wet-lab trials, the design of multi-epitope vaccines is a successful method for combating the majority of infectious diseases, it may be concluded. This demonstrates that this method is one of the most effective ways to manage bacterial and viral infections in commercial fish species.

### Fish vaccine design based on immunoinformatics approach: promising strategies

As we consider the future of immunoinformatics in fish vaccine design, it will be of the utmost importance to cultivate collaboration and knowledge sharing within this field. Several promising strategies can be employed to achieve this goal:

i. The establishment of open-access platforms for researchers to share their data, methodologies, and immunoinformatics tools is essential to fostering knowledge exchange. These platforms could include preprint servers for early dissemination of research findings, data repositories for sharing raw datasets, and open-source bioinformatics software repositories. The collective exchange of information could accelerate progress in this field by minimising duplication of effort and allowing scientists to iterate on previous work.

ii. Promoting international collaborations could be instrumental in combining diverse skills, resources, and points of view. Networks that facilitate collaboration between immunologists, bioinformaticians, and aquaculture specialists, among others, would expedite the integration of diverse ideas and methodologies, thereby fostering innovation in the field.

iii. Introducing capacity-building programmes and workshops in immunoinformatics and related disciplines would equip researchers with the necessary knowledge and skills, particularly those in regions with limited resources. This would enable more scientists to contribute to the field and cultivate a research community that is more globally representative.

iv. Policymakers can provide impetus by creating policies that promote data exchange, collaboration, and research in this field. Furthermore, the allocation of funds specifically for aquaculture immunoinformatics research would stimulate activity and promote innovation in this field.

v. Promoting collaboration between public research institutions and private industry could pool resources and expertise, thereby accelerating the translation of research findings into practical applications, such as novel fish vaccines.

vi. Ethics and Governance: As collaborations expand and research becomes more data-driven, it is crucial to establish robust ethical and governance frameworks. These protocols should include data privacy, intellectual property rights, and equitable benefit-sharing in order to ensure the ethical and efficient execution of collaborations.

Attention should be given to the potential of immunoinformatics and multi-epitope vaccine design to improve vaccine manufacturing capacity, particularly in low- and middle-income countries (LMICs). These computational approaches, which rely on accessible and affordable digital resources rather than costly laboratory infrastructure, have the potential to revolutionise vaccine development in environments with limited resources. Immunoinformatics can reduce the cost of vaccine development, a crucial factor for LMICs. It facilitates the antigen discovery procedure by enabling researchers to predict antigenic components of a pathogen that are likely to elicit a robust immune response using computational methods. This avoids the costly and time-consuming process of empirical experimentation, making the development of vaccines more affordable for institutions with limited resources. Moreover, multi-epitope vaccines, which contain sequences from multiple epitopes, provide additional benefits. They are typically manufactured synthetically, allowing LMICs to potentially produce their own vaccines instead of relying on imports. This factor could result in substantial cost savings and increase local biotechnology expertise.

In addition, the reduced cold chain requirements of multi-epitope vaccines, as a result of their increased thermal stability, could alleviate the logistical challenges associated with vaccine distribution in LMICs. In these regions, where maintaining the necessary low-temperature conditions is often challenging, overcoming cold chain constraints is a pressing necessity. Lastly, the adaptability of multi-epitope vaccine design can pave the way for the development of custom vaccines. These could be modified to target specific pathogen strains prevalent in certain geographic regions, enabling more effective and targeted immunisation strategies. Implementing these strategies could considerably increase collaboration and information exchange in the field of immunoinformatics-based fish vaccine design. By doing so, we can foster a global, collaborative research community with the common objective of developing more effective and sustainable aquaculture health management solutions.

## CONCLUSIONS

Vaccination is fundamental to the sustainable management of aquaculture, playing a crucial role in preventing disease outbreaks and the overuse of antibiotics in fish. However, research and development on vaccines for aquatic animals are still in their infancy, highlighting the pressing need for enhanced strategies. The development of species-specific vaccines requires a comprehensive understanding of fish immune systems, including B-cells, T-cells, MHC molecules, and TLR signalling pathways; however, traditional approaches are often costly and time-consuming. Immunoinformatics is a prospective alternative for more effective vaccine design, capable of addressing the complexities of emerging and re-emerging diseases, antigenic diversity, and personalised immunisation requirements. This strategy employs high-performance tools for identifying multi-epitope vaccines, thereby providing a platform for examining variations in immune adaptations among fish species. However, there are gaps and limitations in the discipline.

Despite the promise of immunoinformatics, the accuracy of these predictions is inextricably linked to the precision of the data and the sophistication of the algorithms
employed. Inaccuracies in high-throughput sequence data highlight the need for robust data cleansing methods and verification protocols; without them, subsequent analyses may be erroneous. Future research should concentrate on refining the computational techniques used in immunoinformatics and developing better methods for integrating these techniques with conventional laboratory testing. This will not only result in a more precise vaccine design based on epitopes, but it will also promote a targeted, cost-effective, and safer approach to fish vaccination. Moreover, the establishment of a comprehensive database of fish immune responses at the individual, species, and population levels could provide valuable insights for vaccine design and delivery strategies. Although immunoinformatics is a promising instrument for vaccine development, it must be continuously refined and validated. In order to push the boundaries of aquatic animal health management in the coming years, it will be essential to adopt this technology and simultaneously address its gaps and limitations.

Considering the undeniable fact that the increase in immunoinformatics studies following COVID-19 has broadened our understanding and presented numerous potential vaccine candidates, the true value of these findings cannot be confirmed without laboratory validation. Computational analysis and *in silico* modelling are essential instruments, but they are only the first step in a multi-step process leading to the development of a viable vaccine. The majority of published immunoinformatics studies are indeed predominantly computational, and their contribution to actual vaccine development can be limited in the absence of experimental validation. The prediction of potential antigens or epitopes is based on our current knowledge of protein structure and immune response, both of which are still active research areas. In addition, immunoinformatics tools are imperfect and frequently operate based on assumptions that may not always be true. Therefore, experimental data must always validate bioinformatics predictions.

Determining the protective efficacy of identified vaccine candidates in aquaculture is difficult due to the complexity of immune responses in aquatic organisms and the difference between their immune systems and those of mammals. Important is experimental validation in the form of laboratory and field evaluations. These include, but are not limited to, epitope mapping, evaluating for immune response in cell cultures or fish, and observing the progression of disease resistance in vaccinated populations. Although the redundancy of theoretical vaccine papers can be a cause for concern, it is essential to remember that these studies still contribute to our collective knowledge and may serve as the foundation for future experimental research. Increasing accessibility to computational tools and techniques has democratised scientific research, allowing more scientists to contribute their findings. As these findings are tested and validated in the laboratory, we will be able to enhance the predictive power of immunoinformatics by refining our tools and models.

To ensure that immunoinformatics research effectively contributes to the development of essential vaccines, it is crucial to encourage the laboratory application of these computational findings. This can be accomplished by encouraging collaboration between computational and experimental biologists, promoting funding for validation studies, and emphasising the publication of studies that include both computational and experimental components. In conclusion, although the influx of theoretical vaccine papers provides a

plethora of potential vaccine candidates, it is essential to take these findings to the next level by conducting the necessary experimental validations to advance vaccine development in the aquaculture field.

### Funding

This work was supported by the Ministry of Higher Education (MoHE) Malaysia through Fundamental Research Grants Scheme (FRGS/1/2021/STG01/UMT/02/2) and Universiti Malaysia Terengganu through the Talent and Publication Enhancement Research Grant (UMT/TAPE-RG/2020/55298). The funders had no role in study design, data collection and analysis, decision to publish, or preparation of the manuscript.

### Grant Disclosures

The following grant information was disclosed by the authors:
Ministry of Higher Education (MoHE) Malaysia: FRGS/1/2021/STG01/UMT/02/2.
Universiti Malaysia Terengganu: UMT/TAPE-RG/2020/55298.

### Competing Interests

The authors declare there are no competing interests.

### Author Contributions

- Siti Aisyah Razali conceived and designed the experiments, performed the experiments, analyzed the data, prepared figures and/or tables, authored or reviewed drafts of the article, funding acquisition; project administration; supervision, and approved the final draft.
- Mohd Shahir Shamsir analyzed the data, authored or reviewed drafts of the article, and approved the final draft.
- Nur Farahin Ishak performed the experiments, authored or reviewed drafts of the article, and approved the final draft.
- Chen-Fei Low analyzed the data, authored or reviewed drafts of the article, and approved the final draft.
- Wan-Atirah Azemin conceived and designed the experiments, performed the experiments, analyzed the data, prepared figures and/or tables, authored or reviewed drafts of the article, and approved the final draft.

### Data Availability

  This is a literature review.

### Supplemental Information

Supplemental information for this article can be found online at http://dx.doi.org/10.7717/peerj.16419#supplemental-information.

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
