# Peer review of "Riding the wave of innovation: immunoinformatics in fish disease control"

_PeerJ, doi:10.7717/peerj.16419_

## Round 0.1 · original submission · Major Revisions

The manuscript needs to be revised thoroughly. Please revise the manuscript based on the reviewers' comments. A point-by-point response letter should be provided when resubmitting the manuscript to the PeerJ for consideration.

·

Basic reporting

This manuscript attempts to describe the applications of using immunoinformatic approaches to improve vaccine design and development for use in aquaculture, specifically, fishes. It first provides an overview of the different types of vaccines, their advantages, and disadvantages, and then proceeds to focus on multi-epitope vaccine design and providing examples of how in-silico methods can be used at every step of the vaccine design process. It also provides case-studies where immunoinformatics has aided in designing multi-epitope vaccines against specific pathogens. Apart from immunoinformatics, it also provides a detailed characterization of the TLR signaling pathways. However, I have some major concerns:
1. The English language used in this paper has a lot of scope for improvement and it would be better to have it written by a person fluent in English.
2. It is not clear how immunoinformatics methods can circumvent the issues related to development of fish vaccines. This aspect needs to be elaborated. Although immunoinformatics can predict epitopes, how does it solve issues specific to development of fish vaccines?
3. I would have liked to read about the differences in the immune system between fishes and humans, in the introduction.
4. The introduction also needs to provide an overview of the common diseases that affect fishes and their causative agents. This could be followed by giving examples of vaccines currently used in fishes and how different they are from vaccines meant for human use.
5. Mentioning something about the measures taken to prevent infection from fishes spreading to humans, might also be worth considering, especially when the boundary between humans and animals keeps getting blurred

Experimental design

While all the steps used for in-silico design of vaccines are mentioned, the authors fail to discuss anything about recent advances in the area of machine learning and deep learning, such as AlphaFold, RosettaFold etc to predict structures of proteins.

The advantages and disadvantages of each method described under 'Construction of multi-epitope vaccines" needs to be stated clearly.

It would be advantageous to write about the use of self-assembling immunogens.

Validity of the findings

The conclusion section needs to be expanded as it does not sufficiently address the future directions and identify potential gaps and possibilities to address those gaps.

·

Basic reporting

No comment

Experimental design

No comment

Validity of the findings

No comment

Additional comments

Reviewer Comments

This review focuses on immunoinformatics software used in recent years for vaccine design, specifically in the context of fish vaccine development. It provides a comprehensive summary of the software and discusses how it is used to design vaccines based on in silico epitopes, develop multi-epitope vaccines, and explore the molecular interactions of immunogenic vaccines. The review examines the practical and effective results of using immunoinformatics to address fish diseases and their frequency. It also predicts immune mechanisms and the application of immunoinformatics in fish diseases using TLR signaling pathways.

1. Please provide more context and background information on the importance of aquaculture in meeting the global demand for fish and the challenges it faces, such as increasing unsustainable fishing practices and the need for disease control.

2. It would be helpful to include specific examples of the economic losses caused by fish disease outbreaks in the aquaculture industry to emphasize the significance of the problem.

3. Please include a statement about the safety and efficacy of fish vaccines in general to address any potential concerns or misconceptions readers may have.

4. Provide more information on the role of immunostimulants and adjuvants in fish vaccines, including their mechanisms of action and specific examples of commonly used substances.

5. When discussing the methods of vaccine administration (injection, immersion, oral), elaborate on the advantages and disadvantages of each method, such as efficacy, practicality, and potential stress on the fish.

6. Include more information on the specific side effects that fish may experience after vaccination and strategies to mitigate these effects.

7. Majority of the B- and T-cell prediction tools were developed and trained using data derived from human and mammalian MHC/HLA alleles. Authors need to explain further how could the tools may address in vaccine design for aquaculture?

8. What is the potential role of immunoinformatics and multi-epitope vaccine design in enhancing vaccine manufacturing capacity in low- and middle-income countries?

9. How can researchers and policymakers support the expansion of collaboration and knowledge exchange in the field of fish vaccine design based on immunoinformatics approach, and what are some promising strategies for achieving this goal?

10. As we are aware there are plenty of immunoinformatics papers published since emergence of COVID-19. However majority of these papers published recently were focusing only on computationally analysis using online tools (without lab-based validation). How do you determine the protective efficacy of the identified vaccine candidates of such papers especially in the area of aquaculture? The redundancy of theoretical vaccine papers could not help much in essential vaccine development. Please comment this in your discussion.

---

## Round 0.2 · accepted · Accept

The authors have thoroughly addressed the reviewers' comments and concerns. This manuscript is acceptable now.

·

Basic reporting

No comment

Experimental design

No comment

Validity of the findings

No comment

Additional comments

No comment